# Interventionally Consistent Surrogates for Complex Simulation Models

**Joel Dyer**[*]
University of Oxford

**Nicholas Bishop**
University of Oxford

**Yorgos Felekis**
University of Warwick

**Fabio Massimo Zennaro**
University of Bergen

**Anisoara Calinescu**
University of Oxford

**Theodoros Damoulas**
University of Warwick

**Michael Wooldridge**
University of Oxford

## Abstract

Large-scale simulation models of complex socio-technical systems provide decision-makers with high-fidelity testbeds in which policy interventions can be evaluated and *what-if* scenarios explored. Unfortunately, the high computational cost of such models inhibits their widespread use in policy-making settings. Surrogate models can address these computational limitations, but to do so they must behave consistently with the simulator under interventions of interest. In this paper, we build upon recent developments in causal abstractions to develop a framework for learning interventionally consistent surrogate models for large-scale, complex simulation models. We provide theoretical results showing that our proposed approach induces surrogates to behave consistently with high probability with respect to the simulator across interventions of interest, facilitating rapid experimentation with policy interventions in complex systems. We further demonstrate with empirical studies that conventionally trained surrogates can misjudge the effect of interventions and misguide decision-makers towards suboptimal interventions, while surrogates trained for *interventional* consistency with our method closely mimic the behaviour of the original simulator under interventions of interest.

## 1 Introduction

Large-scale, complex simulators are powerful tools for modelling distributed socio-technical systems and emergent phenomena across application domains, including the social sciences [Wiese et al., 2024], epidemiology [Kerr et al., 2021], and finance [Cont, 2007]. Many such systems consist of a multitude of autonomous, interacting, and decision-making agents, whose individual behaviours and interactions can be captured more readily and at a higher degree of fidelity in a computer program than through conventional modelling paradigms. This level of granularity can, in turn, allow for more effective control of the potentially deleterious effects that can arise from the endogenous dynamics of real-world systems by providing a testbed for experimentation with policy interventions. In economics, for example, such interventions may take the form of limits on loan-to-value ratios in housing markets to attenuate housing price cycles [Baptista et al., 2016], while in epidemiology they may be (non-)pharmaceutical interventions that aim to inhibit disease transmission [Kerr et al., 2021].

Whilst simulation modelling of this kind promises many benefits, the intricacy and multi-scale nature of the simulators that result from these modelling efforts can result in large computational costs even

---

[*]joel.dyer@cs.ox.ac.uk

38th Conference on Neural Information Processing Systems (NeurIPS 2024).

for single forward simulations [Jagiella et al., 2017, Fadai et al., 2019, Wright and Davidson, 2020, Heppenstall et al., 2021]. Since extensive simulation studies are often required to aid decision-making with these models, such costs present a barrier to their use as synthetic test environments for potential policy interventions in practice. Moreover, the high-fidelity data generated by detailed simulation models can be difficult for decision-makers to interpret and relate to policy interventions that act system-wide [Haldane and Turrell, 2018]. This motivates the development of simpler *surrogate* models that model the underlying system at a higher level of abstraction. Such surrogates can also be used in place of the complex model for downstream tasks where computational resources are limited. In addition, surrogates may be viewed as interpretable explanations for the complex simulator, and they allow for rapid testing of population-wide interventions which may be difficult to implement or test within the orginal model.

However, for surrogates to be useful in downstream tasks involving experimentation with possible policy interventions, they must preserve the complex simulator's dynamics under the external interventions of interest. Without imposing this condition on the constructed surrogate, there is no guarantee that the surrogate will behave similarly under external policy interventions, which in turn may lead policy-makers away from effective policies and towards suboptimal interventions. Existing methods typically apply off-the-shelf machine learning methods to learn surrogates through observation [Lamperti et al., 2018, Platt, 2022], which fails to account for interventional consistency.

**Our contribution.**   To address this, we build on recent developments in *causal abstraction* [Beckers and Halpern, 2019, Zennaro et al., 2023a]. We view the complex simulator and its surrogate as *structural causal models* [Pearl, 2009], and propose a framework for constructing and learning surrogate models for expensive simulators of complex socio-technical systems that are *interventionally consistent*, in the sense that they (approximately) preserve the behaviour of the simulator under equivalent policy interventions. This perspective enables *treating the surrogate model as a causal abstraction of the simulator*. We motivate our proposed methodology theoretically, and demonstrate with simulation studies that our method permits us to learn an abstracted surrogate model for an epidemiological agent-based model that behaves consistently in multiple interventional regimes.

Our approach establishes, for the first time, a connection between complex simulation models and causal abstraction, and a practical approach to learning interventionally consistent surrogates for complex simulators. Our work provides an avenue for researchers modelling complex socio-technical systems to draw on the rich literature in causality for integrating causal knowledge, evaluating *what-if* scenarios, and learning new abstracted models with guarantees about interventional consistency. Our contribution lays the groundwork for surrogate modelling methods that facilitate rapid experimentation with different scenarios and interventions, with assurances that the error introduced by experimenting at a higher level of abstraction is low. This line of work has the potential to enable decision- and policy-makers to use simulation models to quickly identify life-saving policy strategies during novel and rapidly unfolding emergencies, such as pandemics and economic crises. Indeed, a recent World Health Organisation report [World Health Organization et al., 2024] emphasises the importance of integrated modelling to concurrently address interdependent policy objectives, such as reducing disease transmission, mitigating hospital admissions overload, and minimising the economic costs of service closures on society during pandemics. It further discusses the intense time pressures involved in these efforts. Our work addresses these points by taking steps towards facilitating rapid experimentation with large and computationally expensive integrated simulation models.

## 2   Background

We first recall the key elements of causal inference, following Pearl [2009], and elucidate the connection between structural causal models (SCMs) and complex simulators. We also review the notion of exact transformations between SCMs, which theoretically motivates our framework.

### 2.1   Structural causal models

A SCM is a rigorous model describing a causal system:

**Definition 1** (SCMs [Pearl, 2009])**.** *A structural causal model $\mathcal{M}$ is a tuple $\langle \mathbf{X}, \mathbf{U}, \mathcal{F}, \mathbb{P}(\mathbf{U}) \rangle$ where:*

- $\mathbf{X} = \{X_i\}_{i=1}^n$, *is a finite set of endogenous random variables $X_i$ each with domain $\mathrm{dom}[X_i]$;*

- $\mathbf{U} = \{U_i\}_{i=1}^n$, is a finite set of exogenous random variables, each with domain $\mathrm{dom}[U_i]$ and each associated with an endogenous variable;

- $\mathcal{F} = \{f_i\}_{i=1}^n$, is a finite set of measurable structural functions, one for each endogenous variable defined as $f_i : \mathrm{dom}[PA(X_i)] \times \mathrm{dom}[U_i] \to \mathrm{dom}[X_i]$, where $PA(X_i) \subseteq \mathbf{X} \setminus X_i$.

- $\mathbb{P}_{\mathcal{M}}(\mathbf{U})$ is a joint probability distribution over the exogenous variables factorizing as $\prod_{i=1}^n \mathbb{P}_{\mathcal{M}}(U_i)$.

*The model $\mathcal{M}$ is associated with a Directed Acyclic Graph (DAG) $\mathcal{G}_{\mathcal{M}} = \langle \mathcal{V}, \mathcal{E} \rangle$ where the set $\mathcal{V}$ of vertices is given by $\mathbf{X} \cup \mathbf{U}$ and the set $\mathcal{E}$ of edges is given by $\{(S_j, X_i) \mid S_j \in PA(X_i) \cup \{U_i\}\}_{i=1}^n$.*

Definition 1 conforms to the standard definition of a *Markovian SCM* (see Appendix A for an explanation of the underlying assumptions). Thanks to the measurability of the structural functions in $\mathcal{F}$, the probability distribution $\mathbb{P}_{\mathcal{M}}(\mathbf{U})$ over the exogenous variables can be pushed forward over the endogenous variables, defining the probability distribution $\mathbb{P}_{\mathcal{M}}(\mathbf{X}) = \mathcal{F}_{\#}(\mathbb{P}_{\mathcal{M}}(\mathbf{U}))$. Joint distributions $\mathbb{P}_{\mathcal{M}}(\mathbf{S})$ can then be defined for any subset $\mathbf{S} \subseteq \mathbf{X}$.

External interventions on the system by an experimenter can be represented in an SCM through changes in the structural functions. Here, we restrict our attention to hard interventions, in which fixed values are assigned to subsets of endogenous variables:

**Definition 2** (Interventions [Pearl, 2009]). *Given an SCM $\mathcal{M}$, $\mathbf{S} \subseteq \mathbf{X}$, and a set of values $\mathbf{s}$ realizing $\mathbf{S}$, an intervention $\iota = \mathrm{do}(\mathbf{S} = \mathbf{s})$, is an operator that replaces each function $f_i$ associated with $S_i$ with constant $s_i$.*

The intervention $\iota = \mathrm{do}(\mathbf{S} = \mathbf{s})$ induces a new *post-intervention* SCM, $\mathcal{M}_\iota = \langle \mathbf{X}, \mathbf{U}, \mathcal{F}_\iota, \mathbb{P}(\mathbf{U}) \rangle$, identical to the original one, except that in $\mathcal{F}_\iota$ the functions $f_i$ are replaced with the constants $s_i$. The probability distribution of $\mathcal{M}_\iota$ is computed as $\mathbb{P}_{\mathcal{M}_\iota}(\mathbf{X} \setminus \mathbf{S})$. Graphically, the intervention $\iota$ transforms the DAG of $\mathcal{M}$ by removing incoming edges in each variable $S_i$.

We use $\mathcal{I}$ to denote a set of feasible interventions on the SCM $\mathcal{M}$ that are relevant to a policymaker. Intervention sets are equipped with a natural partial ordering: let $\iota_1 = (\mathbf{S} = \mathbf{s})$ and $\iota_2 = (\mathbf{T} = \mathbf{t})$; then $\iota_1 \preceq \iota_2$ iff (i) $\mathbf{S} \subseteq \mathbf{T}$, and (ii) for each $S_i = T_i$ it holds $s_i = t_i$; informally, $\iota_1$ intervenes on a subset of the variables that $\iota_2$ intervenes on, and it sets the same values as $\iota_2$.

## 2.2 Complex simulators as structural causal models

Many simulation models of complex systems – such as, for example, agent-based models (ABMs) – can be modelled as a SCM by expressing its implicit underlying causal structure. Practically, this entails encoding quantities of interest as endogenous variables, deterministic dynamics into structural equations, and factoring sources of randomness into exogenous variables. The following example illustrates how a common ABM from epidemiology can be cast as a SCM.

**Example 1** (Spatial SIRS ABM). *We consider a susceptible-infected-recovered-susceptible (SIRS) epidemic model on an $L \times L$ lattice of cells, each of which represents one of $N = L^2$ agents. The state of each agent can be 0, 1, or 2, respectively, indicating that the agent is disease-free and susceptible to infection, infected, or is recovered from a recent infection. The infection status of all agents at discrete time step $t \in [\![0, T]\!]$ is written as $\mathbf{x}_t \in \{0, 1, 2\}^N$, where $T$ is the total number of simulated time steps, and $[\![l, m]\!] = \{l, l+1, \ldots, m-1, m\}$ for integers $l \leq m$. The states $\mathbf{x}_{t,n}$ of each of the agents $n \in [\![1, N]\!]$ are updated synchronously as follows for $t \in [\![0, T-1]\!]$:*

*(U1) If $\mathbf{x}_{t,n} = 0$, then $\mathbf{x}_{t+1,n} = 1$ with probability*

$$p_{t,n}(\alpha_{t+1}) = 1 - (1 - \alpha_{t+1})^{\sum_{n' \in \mathcal{N}_n} \mathbb{I}[\mathbf{x}_{t,n'}=1]} \tag{1}$$

*where $\mathcal{N}_n$ is the von Neumann neighbourhood for cell $n$; else remain susceptible.*

*(U2) If $\mathbf{x}_{t,n} = 1$, then $\mathbf{x}_{t+1,n} = 2$ with probability $\beta_{t+1}$; else remain infected.*

*(U3) If $\mathbf{x}_{t,n} = 2$, then $\mathbf{x}_{t+1,n} = 0$ with probability $\gamma_{t+1}$; else remain recovered.*

*In the above, $\boldsymbol{\theta}_t = (\alpha_t, \beta_t, \gamma_t) \in [0, 1]^3$ are the model parameters determining the transition probabilities between states. While these may vary over time, the simplest case consists of assigning all $\boldsymbol{\theta}_t$ the same vector,*

$$\boldsymbol{\theta}_t = \mathbf{v} \qquad \forall t \in [\![1, T]\!]. \tag{2}$$

*The model is initialised by infecting each agent in the model at initial time $t = 0$ with probability $I_0 \in [0, 1]$. The value of $I_0$ for any forward simulation of the model can be chosen by drawing a random variable $a$ from some distribution on $[0, 1]$ and setting*

$$I_0 = a. \tag{3}$$

*With this model in place, lockdowns over some time period $t_l : t_l + \Delta$ of length $\Delta \geq 0$ can be modelled (crudely) by setting $\boldsymbol{\theta}_{t_l:t_l+\Delta} = (0, \beta, \gamma)$ for $\beta, \gamma \in [0, 1]$. To express this ABM as an SCM, we define the following:*

**Endogenous variables** *These consist of the variables of interest that may be set by the policymaker: $I_0$, $\{\mathbf{x}_t\}_{0 \leq t \leq T}$, and $\{\boldsymbol{\theta}_t\}_{1 \leq t \leq T}$.*

**Exogenous variables** *The model as described above is initialised randomly according to $a$, $\mathbf{v}$, and a collection $\mathbf{u}_0 = (\mathbf{u}_{0,n})_{1 \leq n \leq N}$ of $N$ random variables distributed as $\mathcal{U}(0, 1)$, the $n$th of which helps determine whether agent $n$ is infected at time $t = 0$. Similarly, further collections $\mathbf{u}_t, t \in [\![1, T]\!]$ of $\mathcal{U}(0, 1)$ random variables decide how each agent updates their state at each time step. Thus the exogenous variables for the model are $a$, $\mathbf{v}$, and the $\mathbf{u}_t$ for $t \in [\![0, T]\!]$.*

**Structural equations** *Equations 2 and 3, respectively, define the structural equations $f_{\boldsymbol{\theta}_t}$ and $f_{I_0}$ for the endogenous variables $\boldsymbol{\theta}_t$ and $I_0$. The structural equation $f_{\mathbf{x}_{0,n}}$ for each $\mathbf{x}_{0,n}, n \in [\![1, N]\!]$ can furthermore be written as*

$$\mathbf{x}_{0,n} = f_{\mathbf{x}_{0,n}}(\mathbf{u}_{0,n}, I_0) = \mathbb{I}\left[\mathbf{u}_{0,n} < I_0\right]. \tag{4}$$

*Finally, update rules (U1)-(U3) can be written in the following way for $t \in [\![0, T-1]\!]$:*

$$
\begin{aligned}
\mathbf{x}_{t+1,n} &= f_{\mathbf{x}_{t+1,n}}(\boldsymbol{\theta}_{t+1}, \mathbf{u}_{t+1,n}, \mathbf{x}_{t,n}) \\
&= \mathbb{I}\left[\mathbf{x}_{t,n} = 0\right] \cdot \mathbb{I}\left[\mathbf{u}_{t+1,n} < p_{t,n}(\alpha_{t+1})\right] + \mathbb{I}\left[\mathbf{x}_{t,n} = 1\right] \cdot (1 + \mathbb{I}\left[\mathbf{u}_{t+1,n} < \beta_{t+1}\right]) \\
&\quad + 2\mathbb{I}\left[\mathbf{x}_{t,n} = 2\right] \cdot (1 - \mathbb{I}\left[\mathbf{u}_{t+1,n} < \gamma_{t+1}\right]),
\end{aligned} \tag{5}
$$

**Distribution over exogenous variables** *The (random) behaviour of the exogenous variables is fully specified by the distribution over $a$ and $\mathbf{v}$, along with $\mathcal{U}(0, 1)$ distributions over the $\mathbf{u}_{t,n}$.*

**The underlying graph** *The DAG corresponding to this SCM is shown in Figure 1 for $T = 3$.*

*In this model, interventions in the form of, for example, lockdowns can be (crudely) modelled by intervening on one or more of the $\boldsymbol{\theta}_t$ as $\mathrm{do}(\boldsymbol{\theta}_t = (0, \beta, \gamma))$ for some $\beta, \gamma \in [0, 1]$, while in the observational regime the $\boldsymbol{\theta}_t$ will all be assigned the same value.*

We emphasise that the above example is intended only to illustrate that complex simulators, such as ABMs, can be seen as SCMs; explicitly representing a given simulator as an SCM as in the example above is not required in the sequel.

### 2.3 Causal abstractions

Beside expressing interventions more rigorously, viewing complex simulation models as SCMs allows one to take advantage of the theory of *causal abstraction* to formalise the relationship between the simulator and its surrogate model. Indeed, causal abstraction provides a framework for relating SCMs representing an identical sys-

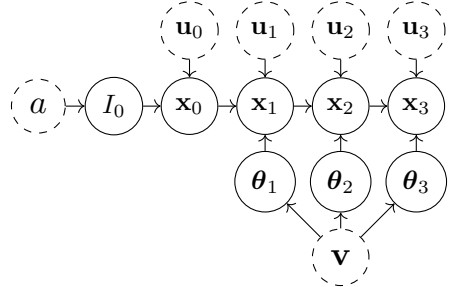

Figure 1: The directed acyclic graph induced by the structural causal model for the spatial SIRS agent-based model for $T = 3$ time steps.

tem at different levels of granularity. The notion of exact transformation formalizes this relation, providing a framework to relate complex models, such as ABMs, to simpler top-down models, while preserving causal structure.

**Definition 3** ($\tau$-$\omega$ Exact Transformation [Rubenstein et al., 2017]). *Given two SCMs, $\mathcal{M}$ and $\mathcal{M}'$, with respective intervention sets $\mathcal{I}$ and $\mathcal{I}'$, a $\tau$-$\omega$ transformation is a pair $(\tau, \omega)$ consisting of a*

*map $\tau : \mathrm{dom}[\mathbf{X}] \to \mathrm{dom}[\mathbf{X}']$ and a surjective, order-preserving map $\omega : \mathcal{I} \to \mathcal{I}'$. An* exact $\tau$-$\omega$ *transformation is a $\tau$-$\omega$ transformation such that*

$$\tau_{\#}(\mathbb{P}_{\mathcal{M}_\iota}) = \mathbb{P}_{\mathcal{M}'_{\omega(\iota)}}, \forall \iota \in \mathcal{I}. \tag{6}$$

An exact $\tau$-$\omega$ transformation constitutes a form of abstraction between probabilistic causal models [Beckers et al., 2020] with the guarantee of commutativity between intervention and transformation as detailed in Figure 2: intervening via $\iota$ and then abstracting produces the same result as abstracting first and then intervening via $\omega(\iota)$. The map $\tau$ describes corresponding states in each of the models, while the map $\omega$ describes corresponding interventions in each model. Whenever the map $\tau$ is clear from context, we herein shorthand the pushforward measure $\tau_{\#}(\mathbb{P}_{\mathcal{M}_\iota})$ as $\mathbb{P}^{\#}_{\mathcal{M}_\iota}$.

An exact $\tau$-$\omega$ transformation between the SCM $\mathcal{M}$ underlying a complex simulation model and the SCM $\mathcal{M}'$ underlying the candidate surrogate model would (a) certify that the surrogate preserves the causal behaviour of interest, guaranteeing interventional consistency when policymakers study interventions through the surrogate, and (b) allow to interpret the emergent causal structure of the simulator through $\mathcal{M}'$.

$$
\begin{array}{ccc}
\iota & \xrightarrow{\ \ \mathcal{M}\ \ } & \mathbb{P}_{\mathcal{M}_\iota} \\
\omega\downarrow & & \downarrow\tau \\
\omega(\iota) & \xrightarrow{\ \ \mathcal{M}'\ \ } & \mathbb{P}_{\mathcal{M}'_{\omega(\iota)}}
\end{array}
$$

Figure 2: Computing $\tau_{\#}(\mathbb{P}_{\mathcal{M}_\iota})$ corresponds to moving right, then down, in the diagram. That is, running the intervention $\iota$ in a base model $\mathcal{M}$ such as an ABM. Computing $\mathbb{P}_{\mathcal{M}'_{\omega(\iota)}}$ corresponds to moving down, then right. That is, running the intervention $\omega(\iota)$ in an abstracted model $\mathcal{M}'$ such as a surrogate. If $(\tau, \omega)$ is an exact transformation, then the diagram is commutative for all interventions. That is, $\tau_{\#}(\mathbb{P}_{\mathcal{M}_\iota}) = \mathbb{P}_{\mathcal{M}'_{\omega(\iota)}}$ for all $\iota \in \mathcal{I}$.

## 3 Abstraction error

It is often unrealistic to assume that an exact $\tau$-$\omega$ transformation exists between a complex simulator and its surrogate A more pragmatic goal is to find an approximate abstraction [Beckers et al., 2020] from the simulator to the surrogate. We therefore define the *abstraction error*:

**Definition 4** (Abstraction error). *Let $(\tau, \omega)$ be a $\tau$-$\omega$ transformation between two SCMs $\mathcal{M}$ and $\mathcal{M}'$ with respective intervention sets $\mathcal{I}$ and $\mathcal{I}'$. Given a statistical divergence $d$ between distributions, and a distribution $\eta$ over the intervention set $\mathcal{I}$, we define the* abstraction error *as follows:*

$$d_{\tau,\omega}(\mathcal{M}, \mathcal{M}') = \mathbb{E}_{\iota \sim \eta}\left[d\left(\tau_{\#}(\mathbb{P}_{\mathcal{M}_\iota}), \mathbb{P}_{\mathcal{M}'_{\omega(\iota)}}\right)\right]. \tag{7}$$

*A $\tau$-$\omega$ transformation is $\alpha$-*approximate *for some $\alpha \in \mathbb{R}_{\geq 0}$ if $d_{\tau,\omega}(\mathcal{M}, \mathcal{M}') \leq \alpha$.*

A $\tau$-$\omega$ abstraction with low abstraction error implies that $\tau_{\#}(\mathbb{P}_{\mathcal{M}_\iota})$ is close to $\mathbb{P}_{\mathcal{M}'_{\omega(\iota)}}$ in expectation with respect to the interventional distribution $\eta$. If the $d\left(\tau_{\#}(\mathbb{P}_{\mathcal{M}_\iota}), \mathbb{P}_{\mathcal{M}'_{\omega(\iota)}}\right)$ is zero for all interventions $\iota \in \mathcal{I}$, then $(\tau, \omega)$ is an exact transformation (see Figure 3).

Definition 4 differs from previously defined notions of abstraction error in the causal abstraction literature. Whilst Beckers et al. [2020] employ a maximum over interventions, we instead take an expectation over a fixed interventional distribution $\eta$. This is motivated by the fact that policymakers will often hold preferences over possible interventions, which may, for example,

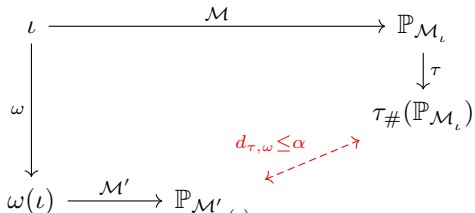

Figure 3: The abstraction error compares the distributions $\tau_{\#}(\mathbb{P}_{\mathcal{M}_\iota})$ and $\mathbb{P}_{\mathcal{M}'_{\omega(\iota)}}$ for each intervention $\iota$ using the divergence $d_{\tau,\omega}$, as indicated by the red dotted arrow. If the divergence is zero then we recover the commutative diagram in Figure 2.

reflect the cost or feasibility of implementing each intervention in the real world. Through the specification of $\eta$, one may implicitly favour surrogates which perform well with respect to interventions of high interest. Further discussion is provided in Appendix B.

## 4 Method

The definitions of abstraction and abstraction error provide us with a framework for learning surrogates, and, in the remainder, we assume that the base model $\mathcal{M}$ is implicitly represented by a

simulation model of a complex socio-technical system. Our goal is then to identify a surrogate model which is interventionally consistent with this simulator. Specifically, from a set of candidate surrogate models $\mathfrak{M}$, we seek a surrogate and a $\tau$-$\omega$ transformation that minimises the abstraction error.

To proceed, we assume that $\mathfrak{M}$ is a parameterised family $\mathfrak{M}^\Psi := \{\mathcal{M}^\psi \ : \ \psi \in \Psi\}$ of differentiable surrogate simulators with tractable probability mass or density function $q^\psi$. Here, $\mathcal{M}^\psi$ denotes the causal model induced by a surrogate whose structural equations are parameterised by $\psi$, and $\Psi$ denotes the set of feasible parameter values. Such a family of surrogate models can be constructed through a composition of differential equation- or deep learning-based modelling, in combination with probability distributions with reparameterisable sampling procedures; an example is a latent neural ordinary differential equation model [Rubanova et al., 2019], which we use in the experiments in Section 5. We further assume only the ability to sample from $\tau_\#(\mathbb{P}_\mathcal{M})$, amounting to running the simulator and applying $\tau$ to the output.

Generally speaking, policymakers know what macroscopic quantities are of interest when modelling a complex system, and how to aggregate the microscopic variables into global statistics. For example, in macroeconomic settings, policymakers will often be concerned with aggregate quantities such as unemployment rates or aggregate demand, which can be derived from the state of the agents. Further specific examples are discussed in Appendix C.1. We thus assume that the map $\tau$, which defines the aggregate, emergent quantities of interest to the policymaker, is pre-specified.

Hence, to find an appropriate $\tau$-$\omega$ transformation, we need only to identify an intervention map $\omega^\star$ between $\mathcal{I}$ and $\mathcal{I}'$. For computational tractability, we select $\omega^\star$ from a parameterised family $\Omega^\Phi := \{\omega^\phi \ : \ \phi \in \Phi\}$ with parameters $\phi$ ranging over the set $\Phi$. For example, $\phi$ may be the weights of a neural network. We then select $\phi^\star$ and $\psi^\star$ jointly by minimising $d_{\tau,\omega}(\mathcal{M}, \mathcal{M}')$ over $\Omega^\Phi \times \mathfrak{M}^\Psi$. Since each element of $\mathfrak{M}$ has a differentiable and tractable distribution, a convenient choice of discrepancy $d$ is the Kullback-Leibler (KL) divergence, such that out problem becomes:

$$\phi^\star, \psi^\star = \underset{\phi \in \Phi, \psi \in \Psi}{\arg\min} \, d_{\tau,\omega^\phi}(\mathcal{M}, \mathcal{M}^\psi) \quad \text{with} \quad d_{\tau,\omega^\phi}(\mathcal{M}, \mathcal{M}^\psi) = \mathbb{E}_\eta \mathbb{E}_{\mathbb{P}_{\mathcal{M}_\iota}^\#} \left[ \log \frac{\mathrm{d}\mathbb{P}_{\mathcal{M}_\iota}^\#}{\mathrm{d}\mathbb{P}_{\mathcal{M}_{\omega^\phi(\iota)}^\psi}} \right]. \quad (8)$$

The KL divergence can be minimised using Monte Carlo estimates of the gradient

$$G(\phi, \psi) = \nabla_{\phi,\psi} \, d_{\tau,\omega^\phi}(\mathcal{M}, \mathcal{M}^\psi) \approx \frac{1}{B} \sum_{b=1}^B -\nabla_{\phi,\psi} \log q_{\omega^\phi(\iota^{(b)})}^\psi(\mathbf{y}^{(b)}) \quad (9)$$

where $\iota^{(b)} \sim \eta$, $\mathbf{y}^{(b)} \sim \tau_\#(\mathbb{P}_{\mathcal{M}_{\iota^{(b)}}})$, $q_{\omega^\phi(\iota)}^\psi$ is the probability mass/density function for $\mathcal{M}_{\omega^\phi(\iota)}^\psi$, and $B \geq 1$ is the size of a batch drawn from $R \geq B$ training examples from the joint distribution over the $\iota^{(b)}$ and $\mathbf{y}^{(b)}$. Once $(\phi^\star, \psi^\star)$ has been selected, we may generate data from the macromodel for ABM intervention $\iota$ by sampling from $\mathbb{P}_{\mathcal{M}_{\omega^{\phi^\star}(\iota)}^{\psi^\star}}$. Algorithm 1 summarises the training procedure.

## 4.1 Theory

Definition 4 is closely related to exact transformations:

**Proposition 1.** *Let $\eta$ be an interventional distribution, $d$ be a statistical divergence, and $(\tau, \omega)$ be a $\tau$-$\omega$ transformation between SCMs $\mathcal{M}$ and $\mathcal{M}'$. If $\tau$-$\omega$ is 0-approximate $(d_{\tau,\omega}(\mathcal{M}, \mathcal{M}') = 0)$, then we have $\eta$-almost-surely*

$$\tau_\#(\mathbb{P}_{\mathcal{M}_\iota}) = \mathbb{P}_{\mathcal{M}'_{\omega(\iota)}}.$$

The proof is in Appendix D.1. In particular, when $\mathcal{I}$ is finite and $\eta(\iota) > 0 \ \forall \iota \in \mathcal{I}$, then any 0-approximate $\tau$-$\omega$ transformation is an exact $\tau$-$\omega$ transformation between $\mathcal{M}$ and $\mathcal{M}'$. This motivates our own choice of loss

---

**Algorithm 1:** Summary of the training procedure.

**Input:** Budget $R$; batch size $B \in [\![1, R-1]\!]$; ABM $\mathcal{M}$; intervention distribution $\eta$; surrogate family $\mathfrak{M}^\Psi$; abstraction map family $\Omega^\Phi$

**Result:** Trained surrogate and abstraction map parameters, $\psi^*$ and $\phi^*$

Set $\mathcal{D} = \emptyset$;
**for** $r = 1$ **to** $R$ **do**
    Sample $\iota^{(r)} \sim \eta$, $\mathbf{x}^{(r)} \sim \mathbb{P}_{\mathcal{M}_{\iota^{(r)}}}$;
    $\mathcal{D} \leftarrow \mathcal{D} \cup (\iota^{(r)}, \tau(\mathbf{x}^{(r)}))$
**end**
**while** *not converged* **do**
    Sample minibatch $\{(\iota^{(b)}, \tau(\mathbf{x}^{(b)}))\}_{b=1}^B$ uniformly from $\mathcal{D}$;
    Take gradient step in $\phi, \psi$ using Equation 9
**end**

function: minimising Equation 8 induces $\phi$ and $\psi$ to produce a surrogate that behaves the same way as the simulator under interventions of interest.

Definition 4 employs an expectation over an interventional distribution $\eta$. As a result, even when the abstraction error is low, there may still be a large discrepancy between $\tau_{\#}(\mathbb{P}_{\mathcal{M}_\iota})$ and $\mathbb{P}_{\mathcal{M}'_{\omega(\iota)}}$ for some fixed intervention $\iota \in \mathcal{I}$. Proposition 2 provides an upper bound on the error associated with any intervention sampled from $\eta$ when $d$ is the KL divergence and the simulator state space is finite:

**Proposition 2.** *Let $d$ be the KL divergence and $CE_\iota = \mathbb{E}_{\mathbf{Y} \sim \tau_{\#}(\mathbb{P}_{\mathcal{M}_\iota})} \left[ -\log q_{\omega(\iota)}(\mathbf{Y}) \right]$ denote the cross-entropy of $\mathbb{P}_{\mathcal{M}'_{\omega(\iota)}}$ with respect to $\tau_{\#}(\mathbb{P}_{\mathcal{M}_\iota})$. Assume $\mathrm{dom}[\mathbf{X}]$ is finite. Then for all $\varepsilon > 0$,*

$$\mathbb{P}_\eta \left( d\left( \tau_{\#}(\mathbb{P}_{\mathcal{M}_\iota}), \, \mathbb{P}_{\mathcal{M}'_{\omega(\iota)}} \right) \geq \varepsilon \right) \leq \frac{\mathbb{E}_{\iota \sim \eta}[CE_\iota]}{\varepsilon}.$$

The proof is in Appendix D.2. This shows that it is only with low probability that the effects of individual interventions are captured poorly by the surrogate when the surrogate and abstraction map parameters, $\psi$ and $\phi$, are found by minimising Equation 8.

## 5 Case study

Here, we outline a case study[2] in which we learn interventionally consistent surrogates for the spatial SIRS ABM from Example 1, allowing us to experiment more rapidly with policy interventions while remaining confident that the causal behaviour of the original SIRS ABM is approximately preserved. Further experimental details and results are given in Appendix E. We consider three families of surrogate models with endogenous variables $\tilde{I}_0 \in [0,1]$, $\tilde{\boldsymbol{\theta}}_t \in \mathbb{R}^3_{\geq 0}$ for $t \in [\![1, T]\!]$, and $\tilde{\mathbf{y}}_t \in \{(a, b, c) \mid a, b, c \in [\![0, N]\!], a + b + c = N\}$ for $t \in [\![0, T]\!]$, where $a, b, c$ denote, respectively, the number of sus-

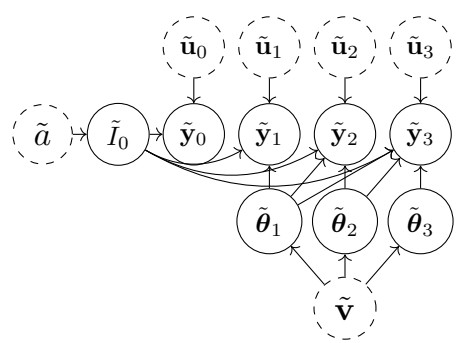

Figure 4: The DAG induced by the SCMs corresponding to the surrogate families for $T = 3$.

ceptible, infected, and recovered individuals in the population. The DAGs underlying the SCMs of each of these families are as in Figure 4, and the three families differ only in the form of the structural equations mapping from $\tilde{I}_0$ and $\tilde{\boldsymbol{\theta}}_{0:t}$ to the $\tilde{\mathbf{y}}_t$. Throughout, we let $q^\psi$ be a Multinomial emission distribution and $\psi$ be trainable parameters of these structural equations.

**Surrogate family 1** consists of a latent ODE (LODE) built by feeding the classical SIRS ODE's three state variables (which take values in the two-simplex) in as the class probabilities of $q^\psi$. Here, $\psi = \emptyset$.

**Surrogate family 2** consists of a latent ODE-RNN (LODE-RNN), where we run a recurrent network (RNN) with parameters $\psi$ over the output of the SIRS ODE. The RNN outputs the class logits of $q^\psi$.

**Surrogate family 3** consists of a latent RNN (LRNN) constructed by running an RNN with trainable parameters $\psi$ over the $\tilde{\boldsymbol{\theta}}_t$, and the output of the RNN at each $t \in [\![0, T]\!]$ indexes the class logits of $q^\psi$.

Given $\tilde{\boldsymbol{\theta}}_{1:T}, \tilde{I}_0$, these surrogates enjoy tractable likelihood functions, which factorise as $q^\psi(\tilde{\mathbf{y}}_{0:T} \mid \tilde{\boldsymbol{\theta}}_{1:T}, \tilde{I}_0) = q^\psi(\tilde{\mathbf{y}}_0 \mid \tilde{I}_0) \prod_{t=1}^{T} q^\psi(\tilde{\mathbf{y}}_t \mid \tilde{\boldsymbol{\theta}}_{1:t}, \tilde{I}_0)$.

**Interventions & the $\tau$-$\omega$ transformation.** Denoting

$$\iota_{\mathbf{v},a} = \mathrm{do}\left( \boldsymbol{\theta}_{1:T} = \mathbf{v}, I_0 = a \right), \tag{10}$$
$$\iota_{\mathbf{v},a,t_l} = \mathrm{do}\left( \boldsymbol{\theta}_{1:t_l-1} = \boldsymbol{\theta}_{t_l+6:T} = \mathbf{v}, \boldsymbol{\theta}_{t_l:t_l+5} = \mathbf{v} \odot (0, 1, 1), I_0 = a \right),$$

we define two subsets $\mathcal{I} = \mathcal{I}_{\mathrm{init}} \cup \mathcal{I}_{\mathrm{init, lock}}$ of interventions for the ABM:

$$\mathcal{I}_{\mathrm{init}} = \{\iota_{\mathbf{v},a} \mid (\mathbf{v}, a) \in [0,1]^4\} \quad \text{and} \quad \mathcal{I}_{\mathrm{init, lock}} = \{\iota_{\mathbf{v},a,t_l} \mid (\mathbf{v}, a, t_l) \in [0,1]^4 \times [\![5, 10]\!]\}. \tag{11}$$

---

[2] Code for reproducing the experimental results is available at `https://github.com/joelnmdyer/neurips_ics4csm`.

Table 1: Metrics for interventionally (**I**) & observationally (**O**) trained surrogates on interventional (**I′**) & observational (**O′**) test sets (median$^{\text{third quartile}}_{\text{first quartile}}$ from 5 repeats). **Bold** denotes best performance.

| Test | Model Train | LRNN | | LODE-RNN | | LODE | |
|---|---|---|---|---|---|---|---|
| | | **I** | **O** | **I** | **O** | **I** | **O** |
| **I′** | AMSE ($\times 10^{-1}$) | $\mathbf{3.48^{3.91}_{3.41}}$ | $49.4^{52.6}_{46.7}$ | $\mathbf{3.35^{3.41}_{3.18}}$ | $18.5^{21.9}_{17.1}$ | $\mathbf{8.15^{8.24}_{8.06}}$ | $22.4^{22.7}_{22.1}$ |
| | ANLL ($\times 10^{3}$) | $\mathbf{2.09^{2.16}_{2.03}}$ | $21.8^{22.9}_{20.1}$ | $\mathbf{1.99^{2.00}_{1.98}}$ | $8.40^{9.89}_{8.20}$ | $\mathbf{4.01^{4.02}_{4.00}}$ | $10.0^{10.1}_{9.91}$ |
| **O′** | AMSE ($\times 10^{-1}$) | $4.13^{4.26}_{4.11}$ | $\mathbf{2.95^{3.16}_{2.62}}$ | $3.59^{3.68}_{3.54}$ | $\mathbf{2.52^{2.78}_{2.16}}$ | $18.4^{18.7}_{18.1}$ | $\mathbf{4.36^{4.40}_{4.32}}$ |
| | ANLL ($\times 10^{3}$) | $2.22^{2.23}_{2.16}$ | $\mathbf{1.64^{1.71}_{1.43}}$ | $1.86^{1.97}_{1.85}$ | $\mathbf{1.43^{1.53}_{1.27}}$ | $7.63^{7.74}_{7.52}$ | $\mathbf{2.15^{2.17}_{2.13}}$ |

The first of these is a subset of interventions that fix the initial proportion of infected individuals in the ABM, as well as its parameter values. The second subset of interventions is the set of interventions that fix (a) the initial proportion of infected individuals in the ABM, (b) the values of the ABM's parameters before, during, and beyond a lockdown beginning at time $t_l \in [\![5, 10]\!]$ with duration equal to 5 time steps, and (c) the value of $t_l$. Similarly defining

$$\iota'_{\tilde{\mathbf{v}},\tilde{a}} = \text{do}\left(\tilde{\boldsymbol{\theta}}_{1:T} = \tilde{\mathbf{v}}, \tilde{I}_0 = \tilde{a}\right), \tag{12}$$

$$\iota'_{\tilde{\mathbf{v}},\tilde{a},\tilde{t}_l} = \text{do}\left(\tilde{\boldsymbol{\theta}}_{1:\tilde{t}_l-1} = \tilde{\boldsymbol{\theta}}_{\tilde{t}_l+6:T} = \tilde{\mathbf{v}}, \tilde{\boldsymbol{\theta}}_{\tilde{t}_l:\tilde{t}_l+5} = \tilde{\mathbf{v}} \odot (0,1,1), = \tilde{\mathbf{v}}, \tilde{I}_0 = \tilde{a}\right),$$

we define $\mathcal{I}' = \mathcal{I}'_{\text{init}} \cup \mathcal{I}'_{\text{init, lock}}$ for the surrogates, where, letting $\mathbb{D} = \mathbb{R}^3_{\geq 0} \times [0,1]$, we have

$$\mathcal{I}'_{\text{init}} = \{\iota'_{\tilde{\mathbf{v}},\tilde{a}} \mid (\tilde{\mathbf{v}}, \tilde{a}) \in \mathbb{D}\} \quad \text{and} \quad \mathcal{I}'_{\text{init, lock}} = \{\iota'_{\tilde{\mathbf{v}},\tilde{a},\tilde{t}_l} \mid (\tilde{\mathbf{v}}, \tilde{a}, \tilde{t}_l) \in \mathbb{D} \times [\![5, 10]\!]\}. \tag{13}$$

The map $\tau$ is taken to map: $\boldsymbol{\theta}_t$ identically to $\tilde{\boldsymbol{\theta}}_t$ for each $t \in [\![1, T]\!]$; the microstate $\mathbf{x}_t$ of the ABM at each time step to the $\tilde{\mathbf{y}}_t$ through an aggregation map that counts the number of agents in $\mathbf{x}_t$ in each of the three states (susceptible, infectious, and recovered); and the initial proportion $I_0$ of infected agents in the ABM identically to $\tilde{I}_0$. Further, for a neural network $f^\phi : [0,1]^3 \to \mathbb{R}^3_{\geq 0}$, we take

$$\omega^\phi : \quad \iota_{\mathbf{v},a} \mapsto \iota'_{f^\phi(\mathbf{v}),a} \quad , \quad \iota_{\mathbf{v},a,t_l} \mapsto \iota'_{f^\phi(\mathbf{v}),a,t_l}. \tag{14}$$

**The benefits of training for interventional consistency** We use Algorithm 1 to jointly learn the parameters $\phi, \psi$ of the surrogates and the map $\omega^\phi$ described above in two different ways: training the surrogate models with $\eta$ taken to be a uniform distribution $\mathcal{U}(\mathcal{I}_{\text{init}})$ over $\mathcal{I}_{\text{init}}$, which entails comparing the behaviour of the surrogate and ABM without lockdowns at different parameters; and training with $\eta$ instead taken to be a uniform distribution $\mathcal{U}(\mathcal{I})$ over $\mathcal{I}$, which entails comparing the behaviour of the surrogate and ABM under different lockdowns, or no lockdowns at all, at different parameters. We indicate the two approaches to training the surrogates with, respectively, bold uppercase **O** and **I**. Appendix E details the training procedure and network architectures. We assess the interventional consistency of the surrogates trained in these two ways by computing error metrics on a hold-out test dataset $\mathbf{I}' = \{(\iota^{(r')}, \mathbf{y}^{(r')}_{0:T})\}^{R'}_{r'=1}$ of size $R' = 1000$, generated as $\iota^{(r')} \sim \eta = \mathcal{U}(\mathcal{I})$, $\mathbf{y}^{(r')}_{0:T} \sim \tau_\#\left(\mathbb{P}_{\mathcal{M}_{\iota^{(r')}}}\right)$. Specifically, we inspect the average mean squared error (AMSE) between trajectories from the trained surrogates and $\mathbf{y}^{(r')}_{0:T}$, and the average negative log-likelihood (ANLL) of this test data under the likelihood of the learned surrogates. Observational consistency is checked on a different hold-out test set $\mathbf{O}'$, generated by instead taking $\eta = \mathcal{U}(\mathcal{I}_{\text{init}})$.

Table 1 shows these performance metrics evaluated on $\mathbf{I}'$ and $\mathbf{O}'$ for all surrogate families and training schemes. We observe that far lower values of the error metrics are obtained by the interventionally, rather than observationally, trained surrogates when assessing interventional consistency. This suggests that training on interventional data can result in more accurate predictions about the effect of interventions in the ABM, and that data drawn from the relevant interventional distributions associated with the ABM should be included during training if the policy-maker intends to perform policy experiments with the surrogate. We also report a minor drop in observational consistency when training with data from the combined intervention set $\mathcal{I}$ instead of $\mathcal{I}_{\text{init}}$, which can be explained by the overfit of the observationally-trained model on the observational distribution. We also observe

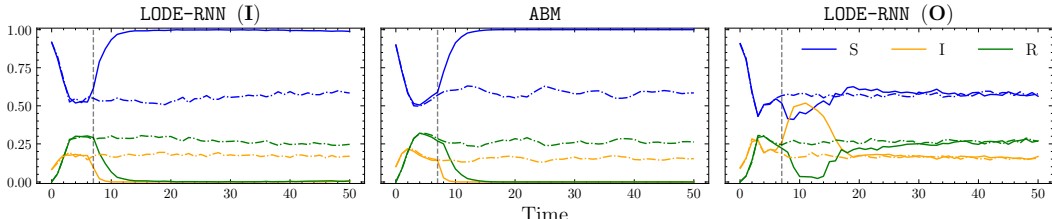

Figure 5: Example trajectories from the `ABM` (middle) and the `LODE-RNN` trained interventionally (left) and observationally (right). A lockdown is imposed at the dashed vertical line. Solid (resp. dot-dash) lines show trajectories under (resp. without) the lockdown. The transmission-inhibiting effect of the lockdown is vastly underestimated in the observationally trained surrogate, while the interventionally trained surrogate accurately predicts a reduction in disease transmission.

that the `LODE-RNN` – which combines the "mechanistic" SIRS `ODE` with a flexible `RNN` – achieves the best interventional and observational consistencies of all surrogates, suggesting that such hybrid approaches to constructing flexible surrogates are promising choices under our proposed method.

In Figure 5, we show an example of a possible negative consequence of failing to train a surrogate on data drawn from the appropriate `ABM` interventional distributions. In the middle panel, we show the change in the `ABM` trajectory induced by imposing a lockdown at time $t_l = 7$, while in the left (resp. right) panel we show corresponding trajectories from the interventionally (resp. observationally) trained surrogates under the equivalent intervention learned through our training procedure. While the interventionally trained `LODE-RNN` correctly predicts that the lockdown effectively impedes the spread of the disease in the `ABM`, the observationally trained surrogate predicts that the lockdown will temporarily *increase* infections, before approximately reverting to the behaviour of the model without a lockdown.

The use of such a surrogate model in policy experiments when limited computational resources do not permit use of the accurate, high-fidelity `ABM` of the underlying complex system may therefore have misdirected policy-makers towards suboptimal, and away from effective, interventions. Indeed, while the SIRS `ABM` predicts that any lockdown is better than no lockdown at all for reducing the number of infections occurring over the simulated time horizon, we see that the observationally trained surrogates often do not predict that no lockdown is the worst intervention in this respect, and in some cases mistakenly predict that no lockdown is the *best* intervention. For example, the observational LRNN predicts that no lockdown was the best intervention in 1 of 5 training repeats, and was not the worst option in all 5 of 5 training repeats. In contrast, none of the interventionally trained surrogates predict that no lockdown is the best intervention, and only the interventional `LODE` model predicts that no lockdown is not the worst option (in only 2 out of 5 training repeats). This highlights the potential importance of training surrogate models for interventional consistency when their purpose is to help inform downstream decision-making tasks. Furthermore, this suggests that a possible benchmark criterion in further research on interventional surrogates could be the degree to which different surrogates preserve the ordering of interventions with respect to those downstream tasks of interest.

## 6  Related work

Surrogates are often used to expedite simulation-based inference when modelling complex systems [Heppenstall et al., 2021]. Modern approaches rely on established machine learning methods such as random forests [Lamperti et al., 2018, De Leeuw et al., 2023], artificial neural networks [Anirudh et al., 2022, De Leeuw et al., 2023], support vector machines [ten Broeke et al., 2021], kriging [Salle and Yıldızoğlu, 2014], and mixture density networks (`MDN`s) [Platt, 2022]. Our experiments also rely on established machine learning methods to construct surrogates; for example, our LRNN surrogate family resembles that of Platt [2022], in which `MDN`s are used to approximate an `ABM`'s transition density. However, in such works, the *causal/interventional* consistency of the surrogate with respect to the simulator and policy interventions of interest is not considered. In contrast to prior work, our work explicitly details the causal relation between the surrogate and the underlying simulator via

causal abstraction, which broadens the scope of surrogate modelling beyond its current use case of expediting calibration to also enable the use of surrogates for policy experimentation.

Causal abstraction and exact transformations were introduced by Rubenstein et al. [2017]. Beckers and Halpern [2019] extended this work by proposing stricter definitions of causal abstraction, and in Beckers et al. [2020], where approximate abstractions are introduced to account for uncertainty and simplification. Causal abstraction found practical application in Geiger et al. [2021] for learning interpretable neural networks. Rischel and Weichwald [2021] discusses an alternative category-theoretical definition of abstraction; this was used to learn abstractions to transfer data between models at different levels of abstraction in Zennaro et al. [2023a]. Further related work includes a multi-marginal Optimal Transport solution to the abstraction learning problem [Felekis et al., 2024], as well as constructive abstraction learning in neural causal models [Xia et al., 2023] and cluster `DAG`s [Anand et al., 2023]. However, none of these approaches reduce the state space of the `SCM` or the cost of simulation, as our approach does.

# 7 Conclusion

We propose a rigorous framework for learning interventionally consistent surrogates for complex simulation models, formalised with casual abstraction. This is the first application of causal abstraction to surrogate modelling. Our approach applies to any simulator corresponding to any `DAG`, and does not require explicit knowledge of the simulator's `SCM`. Through experiments, we highlight the efficacy of our method against purely observational surrogates that do not learn to match interventional data under equivalent interventions. Using our framework, policy-makers may be able to more rapidly draw insights from complex simulators about the possible effects of interventions – in our experiments, our surrogates simulate approximately three times faster than the original complex simulators – and swiftly prepare effective responses to future crises.

Our work naturally suffers limitations. Investigating the sample complexity of abstraction learning would be desirable in future work. Our definition of abstraction error involves an expectation over interventions rather than a maximum as in Beckers et al. [2020]; this produces a computationally tractable optimisation problem, but introduces the possibility that one or more interventions is captured poorly by the learned abstraction map, even for a low abstraction error. In our experiments, we have assumed surrogate models with tractable and differentiable density functions, permitting us to use a KL divergence within our definition of abstraction error; future work might extend our approach by considering different surrogate families with these properties, such as families based on normalising flows [Tabak and Vanden-Eijnden, 2010], or alternative divergences that relax the requirement for tractable densities, such as maximum mean discrepancies [Gretton et al., 2012]. Finally, our method does not directly exploit knowledge of the simulators' causal graphs to accelerate abstraction learning. It is possible that exploiting access to the base `SCM`/`DAG` may expedite abstraction by allowing us to focus on minimal intervention sets [Aglietti et al., 2020, Lee and Bareinboim, 2018], or leverage the identifiability of interventional distributions to reduce the number of simulations required from the base model [Lattimore et al., 2016, Bilodeau et al., 2022]. However, it is unclear whether or not applying the do-calculus on large causal graphs is more efficient than simulating interventions directly. The "black-box" nature of our approach may be beneficial for this reason, and since it does not require the modeller to explicitly write their simulator as an `SCM`, making it generically applicable.

## Acknowledgments and Disclosure of Funding

JD, NB, AC, and MW acknowledge funding from a UKRI AI World Leading Researcher Fellowship awarded to Wooldridge (grant EP/W002949/1). MW and AC also acknowledge funding from Trustworthy AI - Integrating Learning, Optimisation and Reasoning (TAILOR), a project funded by European Union Horizon2020 research and innovation program under Grant Agreement 952215. YF: This scientific paper was supported by the Onassis Foundation - Scholarship ID: F ZR 063-1/2021-2022. TD acknowledges support from a UKRI Turing AI Acceleration Fellowship [EP/V02678X/1].

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

# A Assumptions Underlying Markovian SCMs

Definition 1 implies the standard assumptions of (i) *acyclicity* of the DAG $\mathcal{G}_\mathcal{M}$ and (ii) *causal sufficiency*, meaning that there are no unobserved confounders [Pearl, 2009, Peters et al., 2017]. These two assumptions entail that our SCMs are Markovian.

We also assume *faithfulness*, guaranteeing that independencies in the data are captured in the graphical model Spirtes et al. [2000].

# B Other Notions of Abstraction Error

As discussed in Section 3, Definition 4 is closely related to the notion of abstraction error introduced by Beckers et al. [2020]. In contrast to Definition 4, Beckers et al. [2020] employ a maximum over the intervention set $\mathcal{I}$ instead of an expectation. Hence, the abstraction error introduced by Beckers et al. [2020] may be viewed as a worst-case version of Definition 4.

In addition, Beckers and Halpern [2019] assume the intervention map $\omega$ can be implicitly defined by the map $\tau$, and require the abstraction map $\tau$ to be consistent. That is, the image of $\mathcal{I}$ under the intervention map induced by $\tau$ must equal $\mathcal{I}'$. Since we do not couple the maps $\tau$ and $\omega$ we enforce no such condition. Additionally, Beckers et al. [2020] enforce surjectivity of $\tau$. Since this makes no practical difference in a surrogate's use in downstream tasks, we dispense with this assumption.

Alternative notions of abstraction error have been introduced by Zennaro et al. [2023b], building upon the notion of exact transformations introduced by Rischel [2020]. Such alternative notions of abstraction have recently been used to define transfer learning protocols between SCMs related by an approximate causal abstraction in Zennaro et al. [2024]. We conjecture that an analogous version of our framework may be developed for this setting, wherein the aggregation function over the intervention set is again chosen to be an expectation over an interventional distribution $\eta$ instead of a maximum, and we leave this as a direction for future work.

# C Additional Related Work

Surrogate modelling of complex simulators is closely related to the problem of simulation-based inference. Inference involves tuning model parameters so that data generated by the simulator matches that generated by the real world system being modelled. Analogously, surrogate modelling consists of tuning surrogate parameters so that data generated by the surrogate matches data generated by the corresponding simulator. Hence, methods for calibration can naturally be applied to learn surrogates. Several calibration techniques and metrics have been proposed in the literature, including the method of simulated moments [Fabretti, 2013, Gilli and Winker, 2003] and minimum simulation distance [Grazzini and Richiardi, 2015]. We refer the reader to Dyer et al. [2022, 2024] for thorough surveys. Unlike our framework, surrogates trained for the purpose of parameter estimation do not typically account for interventional consistency explicitly.

More generally, our framework bears similarities to latent space modelling of Markov decision processes (MDPs) [Gelada et al., 2019], wherein one attempts to learn a smaller latent MDP from a target MDP, whose size precludes its use in downstream tasks. For downstream tasks such as formal verification of policies, Delgrange et al. [2022] employs the bisimulation metric to measure the consistency of their latent MDPs with respect to the target. Abstraction error plays an analogous role in our framework, where the original MDP corresponds to the simulator, and the latent MDP the surrogate. Likewise, the surrogates we propose in Section 5 are implicitly connected to the scientific modelling framework of Rackauckas et al. [2021], who embed prior information regarding system dynamics into systems of universal differential equations represented by neural architectures such as neural ODEs. We embed the underlying dynamics of the classical SIRS ODE into several surrogates in an attempt to learn better causal abstractions. Our work also bears some similarities to, yet differs substantially in several key ways from, Kekić et al. [2023], who also use an abstraction error to learn reduced causal models from larger SCMs. While their approach focuses on a single target variable at a fixed time horizon, assumes Gaussian noise and linear structural functions, and focuses on explainability of outcomes, we track multiple interdependent variables over the entire time horizon with a focus on accurate simulation from interventional distributions. Our approach

is therefore more tailored to large-scale and realistic nonlinear simulators. In contrast, the method presented in Kekić et al. [2023] becomes impractical for large-scale models.

## C.1 Examples of $\tau$ Maps in Real Modelling Scenarios

We provide here some practical examples, beyond the two case studies we present, that illustrate how the $\tau$ map may be chosen for large-scale simulators, as a guideline for practitioners. We consider three exaples from the literature on policy modelling below:

1. Consider the model of forced migration in Ghorbani et al. [2024]. Variables of interest to these modellers are the total number of displaced people by location over time by age, gender, and other demographic characteristics. $\tau$ would therefore be defined by counting the number of agents in each of these states at each location, i.e. $\tau_{l,d}(x_t) = \sum_{a \in A} \mathbb{I}[\text{agent } a \text{ has demographic features } d \text{ and is in location } l \text{ at time } t]$ where $l$ indexes locations, $d$ are demographic features, $x$ is the state of the simulation at time $t$, $A$ is the set of all agents, and $\mathbb{I}$ is the indicator function.

2. Consider the model of flood risk mitigation behaviours proposed in Geaves et al. [2024], which models how households decide to take precautions to protect themselves from floods in high flood risk areas. The modellers are interested in the different precautions households take under different policy interventions, namely whether they: do nothing; purchase insurance; purchase property-level protection; and purchase property-level protection and insurance (see Fig. 3 of Geaves et al. [2024]). $\tau$ would count the number of agents taking such actions in this case (as in the example above).

3. Consider the UK housing market model proposed in Bardoscia et al. [2024], in which households consume goods, provide labour and invest in housing, whilst banks assess the credit worthiness of borrowers and set commercial interest rates. Tables 2-6 of Bardoscia et al. [2024] define macroeconomic market statistics such as inflation rate, unemployment rate and real interest rate that are of interest to the modellers. $\tau$ would therefore be defined by standard macroeconomic formulas for these quantities.

# D Proof

## D.1 Proof of Proposition 1

*Proof.* By non-negativity of the divergence $d$ we have $d\left(\tau_{\#}(\mathbb{P}_{\mathcal{M}_\iota}), \mathbb{P}_{\mathcal{M}'_{\omega(\iota)}}\right) \geq 0$ for all $\iota \in \mathcal{I}$. Hence $d_{\tau,\omega}(\mathcal{M}, \mathcal{M}')$ corresponds to an expectation over a non-negative random variable. Since this expectation is equal to zero, we conclude that $d\left(\tau_{\#}(\mathbb{P}_{\mathcal{M}_\iota}), \mathbb{P}_{\mathcal{M}'_{\omega(\iota)}}\right) = 0$ almost surely with respect to the distribution $\eta$. Positivity of the divergence $d$ then implies that $\tau_{\#}(\mathbb{P}_{\mathcal{M}_\iota}) = \mathbb{P}_{\mathcal{M}'_{\omega(\iota)}}$ almost surely with respect to the distribution $\eta$. $\qquad\square$

## D.2 Proof of Proposition 2

*Proof.* Using Markov's inequality and the fact that $d_{\tau,\omega^\phi}(\mathcal{M}, \mathcal{M}^\psi) = \mathbb{E}_{\iota \sim \eta}\left[d\left(\tau_{\#}(\mathbb{P}_{\mathcal{M}_\iota}), \mathbb{P}_{\mathcal{M}^\psi_{\omega^\phi(\iota)}}\right)\right]$:

$$\mathbb{P}_\eta\left(d\left(\tau_{\#}(\mathbb{P}_{\mathcal{M}_\iota}) \,\|\, \mathbb{P}_{\mathcal{M}^\psi_{\omega^\phi(\iota)}}\right) \geq \epsilon\right) \leq \frac{d_{\tau,\omega^\phi}\left(\mathcal{M}, \mathcal{M}^\psi\right)}{\epsilon}.$$

Since we have a finite domain, the likelihood functions associated with (a) the pushforward measure of the ABM under $\tau$ and (b) the surrogate macromodel can be written as probability mass functions, whose logarithms are non-positive. Since we have assumed $\mathbb{P}_{\mathcal{M}^\psi_{\omega^\phi(\iota)}} \ll \tau_{\#}(\mathbb{P}_{\mathcal{M}_\iota})$, we have that

$0 \leq -\log q^\psi_{\omega^\phi(\iota)}(\mathbf{Y}) < \infty$ for any $\mathbf{Y} \sim \tau_{\#}(\mathbb{P}_{\mathcal{M}_\iota})$, and therefore

$$0 \leq \mathbb{E}_{\iota \sim \eta}\left[\mathtt{CE}_\iota\right] < \infty. \tag{15}$$

Figure 6: A schematic representation of the LODE surrogate family for a single time step. First, the output of the SIRS ODE for the next time step, $\mathbf{z}_{t+1}$, is computed via ODESolve. Then, $\mathbf{z}_{t+1}$ serves as the logits for a multinomial distribution from which $\tilde{\mathbf{y}}_t$ is sampled. This sampling procedure is denoted by MN in the diagram. The exogenous variables required to reparameterise the multinomial distribution during sampling are denoted by $\tilde{\mathbf{u}}_t$.

We also have that

$$-\mathbb{H}_{\tau_\#(\mathbb{P}_{\mathcal{M}_\iota})} \leq 0 \ \Rightarrow \ \mathbb{E}_{\iota\sim\eta}\left[-\mathbb{H}_{\tau_\#(\mathbb{P}_{\mathcal{M}_\iota})}\right] \leq 0, \tag{16}$$

where $\mathbb{H}_{\tau_\#(\mathbb{P}_{\mathcal{M}_\iota})}$ is the entropy of the probability mass function associated with $\tau_\#(\mathbb{P}_{\mathcal{M}_\iota})$, and that

$$d\left(\tau_\#(\mathbb{P}_{\mathcal{M}_\iota}), \mathbb{P}_{\mathcal{M}^\psi_{\omega^\phi(\iota)}}\right) = -\mathbb{H}_{\tau_\#(\mathbb{P}_{\mathcal{M}_\iota})} + \mathrm{CE}_\iota \geq 0 \tag{17}$$

$$\Rightarrow d_{\tau,\omega^\phi}\left(\mathcal{M}, \mathcal{M}^\psi\right) = \mathbb{E}_{\iota\sim\eta}\left[-\mathbb{H}_{\tau_\#(\mathbb{P}_{\mathcal{M}_\iota})}\right] + \mathbb{E}_{\iota\sim\eta}\left[\mathrm{CE}_\iota\right] \leq \mathbb{E}_{\iota\sim\eta}\left[\mathrm{CE}_\iota\right]. \tag{18}$$

We write the upper bound above in terms of the cross-entropy, since this can be estimated with finite samples, whereas the full KL-divergence cannot be estimated in general due to the complexity of evaluating the density associated with $\tau_\#(\mathbb{P}_{\mathcal{M}_\iota})$ for an arbitrary ABMs. Hence

$$\mathbb{P}_\eta\left(d\left(\tau_\#(\mathbb{P}_{\mathcal{M}_\iota}) \| \mathbb{P}_{\mathcal{M}^\psi_{\omega^\phi(\iota)}}\right) \geq \epsilon\right) \leq \frac{\mathbb{E}_{\iota\sim\eta}\left[\mathrm{CE}_\iota\right]}{\epsilon}. \tag{19}$$

$\square$

# E   Further Experimental Details

As described in the main text, the three surrogate families we consider have SCMs whose corresponding DAGs can be drawn as in Figure 4. In this section, we fully specify the corresponding SCM for each surrogate. Furthermore, for each surrogate, we provide details on the procedure used to train the parameters $\psi$ and $\phi$, which respectively describe the structural equations of each SCM and their corresponding intervention map $\omega$.

## E.1   The LODE Surrogate Family

To construct a set $\mathfrak{M}$ of probabilistic SCMs, we define a latent neural ordinary differential equation (LNODE) based on the classical SIRS ODE system. The SIRS ODE system takes the form

$$\frac{\mathrm{d}\tilde{S}_t}{\mathrm{d}t} = \tilde{\gamma}_t\tilde{R}_t - \tilde{\alpha}_t\tilde{I}_t\tilde{S}_t, \qquad \frac{\mathrm{d}\tilde{I}_t}{\mathrm{d}t} = \tilde{\alpha}_t\tilde{I}_t\tilde{S}_t - \tilde{\beta}_t\tilde{I}_t,$$
$$\frac{\mathrm{d}\tilde{R}_t}{\mathrm{d}t} = \tilde{\beta}_t\tilde{I}_t - \tilde{\gamma}_t\tilde{R}_t, \tag{20}$$

where $\tilde{\boldsymbol{\theta}}_t = (\tilde{\alpha}_t, \tilde{\beta}_t, \tilde{\gamma}_t) \in \mathbb{R}^3_{\geq 0}$ are the ODE parameters and $\mathbf{z}_t = (\tilde{S}_t, \tilde{I}_t, \tilde{R}_t) \in \mathcal{S} \ \forall t \in [0, T]$ is the ODE state, where $\mathcal{S}$ is the two-simplex. Note that $\mathbf{z}_t$ represents the proportion of susceptible, infected and recovered individuals in the population according the SIRS ODE. Whilst the parameters $\tilde{\boldsymbol{\theta}}_t$ may change over time – which will permit the experimenter to intervene on the values of the parameters at different time steps – we assume the simplest case of assigning the same vector $\tilde{\mathbf{v}} \in \mathbb{R}^3_{\geq 0}$ to all $\tilde{\boldsymbol{\theta}}_t$ when no interventions are applied:

$$\tilde{\boldsymbol{\theta}}_t = \tilde{\mathbf{v}}, \quad \forall t \in [0, T]. \tag{21}$$

In other words, Equation (21) describes the structural equation $\tilde{f}_{\tilde{\boldsymbol{\theta}}_t}$ for $\tilde{\boldsymbol{\theta}}_t$. Practically speaking, the choice of $\tilde{\mathbf{v}}$ is inconsequential, as we can model any change to $\tilde{\boldsymbol{\theta}}_t$ as an intervention. Given $\tilde{\boldsymbol{\theta}}_t$, the ODE state $\mathbf{z}_t$ evolves according to the following rule:

$$\mathbf{z}_t = \mathrm{ODESolve}(\mathbf{z}_{t-1}, \tilde{\boldsymbol{\theta}}_t), \quad t \in [\![1, T]\!], \tag{22}$$

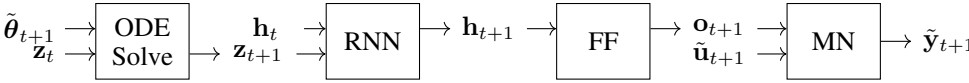

Figure 7: A schematic representation of the `LODE-RNN` surrogate family for a single time step. First, the output of the SIRS `ODE` for the next time step, $\mathbf{z}_{t+1}$, is computed via ODESolve. Then, $\mathbf{z}_{t+1}$ is passed through to the hidden state of a recurrent neural network (denoted by RNN in the diagram) that updates its hidden state from $\mathbf{h}_t$ to $\mathbf{h}_{t+1}$. The updated hidden state is passed to a feedforward neural network (denoted by FF in the diagram), which computes the logits $\mathbf{o}_{t+1}$ for a multinomial distribution from which $\tilde{\mathbf{y}}_{t+1}$ is sampled.

where ODESolve denotes numerical integration of System 20 between times $t - 1$ and $t$. In our experiments, we compute this using a Euler scheme with step size $\Delta t = 1$. The initial state of the `ODE` is taken to be $\mathbf{z}_0 = (1 - \tilde{I}_0, \tilde{I}_0, 0)$. One may change the initial state $\mathbf{z}_0$ through interventions on $\tilde{I}_0$, which is modelled as an endogenous variable.

Given $\mathbf{z}_t$, we draw the endogenous variables $\tilde{\mathbf{y}}_t$ from a multinomial distribution whose class probabilities are given by $\mathbf{z}_t$. Whilst $\mathbf{z}_t$ represents the percentage of susceptible, infected, and recovered individuals predicted by the SIR `ODE`, $\tilde{\mathbf{y}}_t$ represents the actual counts observed by the experimenter. We write $\tilde{f}_{\tilde{\mathbf{y}}_t}(\tilde{I}_0, \tilde{\boldsymbol{\theta}}_{1:t}, \tilde{\mathbf{u}}_t)$ to denote the structural function associated with $\tilde{\mathbf{y}}_t$, where the dependence on $\tilde{I}_0$ and $\tilde{\boldsymbol{\theta}}_{t'}$ for $t' \leq t$ is mediated by the trajectory followed by the $\mathbf{z}_{t'}$ for $t' \leq t$, and $\tilde{\mathbf{u}}_t$ are the exogenous random variables required to reparameterise the multinomial sampling procedure on each time step.

Note that $\psi = \emptyset$ for this family of surrogates, and hence $\mathfrak{M}$ is a singleton. For the function $f^\phi$ comprising the intervention map $\omega^\phi$, we take a feedforward network with layer sizes 3, 32, 64, 64, 64, 32, 3. A ReLU activation is applied after each hidden layer, and a sigmoid activation is applied to the final output layer. The sigmoid activation function ensures that the predicted intervention vector $f^\phi(\mathbf{v})$ on the parameters of the LODE has all of its components in the range $[0, 1]$, which is suitable when forward simulating the `ODE` with an Euler scheme with $\Delta t = 1$. This feedforward network consists of 12,739 trainable parameters.

### E.2 The `LODE-RNN` Surrogate Family

This surrogate family closely mimics the `LODE` family described above, and differs only in that the class *logits* of the multinomial distributions are instead indexed by the output of a feedforward network – with layer sizes 32, 32, 64, 32, 16, 3, where all hidden layers are followed by a ReLU activation function – which maps from the hidden state $\mathbf{h}_t \in \mathbb{R}^{32}$ of a GRU recurrent network that is passed over the trajectory $\mathbf{z}_{0:T}$ generated from the SIRS `ODE` (forward simulated as described above). The combined action of the `ODE` solver, the GRU-feedforward networks, and the reparameterisation of sampling from the multinomial distributions, define the structural equations $\tilde{f}_{\tilde{\mathbf{y}}_t} : (\tilde{I}_0, \tilde{\boldsymbol{\theta}}_{1:t}, \tilde{\mathbf{u}}_t) \mapsto \tilde{\mathbf{y}}_t$ for each $t \in [\![1, T]\!]$.

For this model, $\psi$ is the collection of trainable parameters comprising these GRU and feedforward networks. For $f^\phi$, we use a feedforward network with layer sizes 3, 32, 64, 32, 3, where a ReLU activation is applied after all hidden layers and a sigmoid activation is applied after the final layer. Thus, the total number of trainable parameters from $\psi$ and $\phi$ combined is 13,798.

### E.3 The `LRNN` Surrogate Family

This surrogate family makes no use of the SIRS `ODE` model. Instead, the logits of the multinomial distributions for $t \in [\![1, T]\!]$ are indexed by the outputs $(\mathbf{o}_1, \ldots, \mathbf{o}_T)$, $\mathbf{o}_t \in \mathbb{R}^3$ of a feedforward network – with layer sizes 32, 32, 64, 32, 16, 3, and where all hidden layers are followed by a ReLU activation function – that maps from the hidden state $\mathbf{h}_t \in \mathbb{R}^{32}$ of a GRU recurrent network which is passed over the sequence $\tilde{\boldsymbol{\theta}}_{1:T}$. The initial hidden state is chosen to be $\mathbf{h}_0 = (1 - \tilde{I}_0, \tilde{I}_0, \mathbf{0})$, where $\mathbf{0}$ is a vector of 30 zeros. We also take $\mathbf{o}_0 = (\log(1 - \tilde{I}_0), \log(\tilde{I}_0), -\infty)$ which indexes the logits of the multinomial distribution at time $t = 0$. Once again, we may write the structural equations $\tilde{f}_{\tilde{\mathbf{y}}_t}$ for

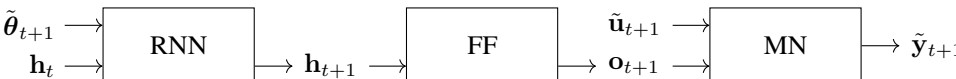

Figure 8: A schematic representation of the LRNN surrogate family for a single time step. First, the parameters $\boldsymbol{\theta}_{t+1}$ are passed to a recurrent neural network (denoted by RNN in the diagram) that updates its hidden state. The updated hidden state is passed to a feedforward neural network (denoted by FF in the diagram), which computes the logits $\mathbf{o}_{t+1}$ for a multinomial distribution from which $\tilde{\mathbf{y}}_{t+1}$ is sampled.

the $\tilde{\mathbf{y}}_t$ in terms of $\tilde{I}_0$, $\tilde{\boldsymbol{\theta}}_{1:t}$, and the exogenous random variables $\tilde{\mathbf{u}}_t$ required to reparameterise the sampling procedure from the multinomial distribution.

Since we use exactly the same networks in this surrogate family as in the LODE-RNN family, the total number of trainable parameters from $\psi$ and $\phi$ combined is also 13,798.

### E.4 The likelihood function for each of these surrogate families

Having intervened on the $\tilde{I}_0$ and $\tilde{\boldsymbol{\theta}}_t$ with known values, the class probabilities for each multinomial distribution is completely determined given the deterministic dynamics within the structural equations mapping to the $\tilde{\mathbf{y}}_t$.

### E.5 Formalising the $\tau$ map

Taking $\mathrm{dom}[I_0] = \mathcal{J}_{\mathcal{M}} = [0,1]$, $\mathrm{dom}[\mathbf{X}_{0:T}] = \mathcal{X}^{T+1}$ with $\mathcal{X} = \{0,1,2\}^N$, and $\mathrm{dom}[\boldsymbol{\Theta}_{1:T}] = \mathcal{P}_{\mathcal{M}}^T$ with $\mathcal{P} = [0,1]^3$, we define

$$\tau : \mathcal{J}_{\mathcal{M}} \times \mathcal{X}^{T+1} \times \mathcal{P}_{\mathcal{M}}^T \to \mathcal{J}_{\mathcal{M}'} \times \mathcal{Y}^{T+1} \times \mathcal{P}_{\mathcal{M}'}^T$$

which operates componentwise as

$$\tau\left(I_0, \mathbf{x}_{0:T}, \boldsymbol{\theta}_{0:T}\right) = \left(\tau_i(I_0), \tau_x(\mathbf{x}_{0:T}), \tau_\theta(\boldsymbol{\theta}_{0:T})\right) \tag{23}$$

where

$$\tau_i = \mathrm{id}, \tag{24}$$

$$\tau_x : \mathbf{x}_{0:T} \mapsto \left(\sum_{n=1}^N \mathbb{I}_{\mathbf{x}_{nt}=0}, \sum_{n=1}^N \mathbb{I}_{\mathbf{x}_{nt}=1}, \sum_{n=1}^N \mathbb{I}_{\mathbf{x}_{nt}=2}\right)_{0:T},$$

$$\tau_\theta = \mathrm{id}. \tag{25}$$

In the above, $\mathrm{id}$ is the identity map, and $\tau_x$ acts by counting the total number of susceptible, infected, and recovered individuals in the ABM at each time step.

### E.6 Further experimental details on the training procedure

All models were trained on CPU on a 2022 MacBook Pro, operating on macOS Ventura 13.2.1. Training one surrogate model on this machine took on average approximately 20 minutes, amounting to approximately 600 minutes in total to produce the results reported in Table 1. Initial attempts at experiments while the code was still in development contribute approximately 200 additional minutes. Software dependencies are specified in the GitHub repository containing the code for this paper, which will be made public upon acceptance.

We assume periodic boundary conditions in both spatial dimensions for the ABM presented in Example 1, which is used in all of our experiments.

As suggested by Figures 1, 4, and 6-8, the parameters $\boldsymbol{\theta}$ and $\tilde{\boldsymbol{\theta}}$ are fed into the models at each time step.

For the LODE and LODE-RNN surrogate families, we forward simulate the SIRS ODE with an Euler scheme with step size $\Delta t = 1$.

For all surrogates, the neural networks comprising the $\omega^\phi$ map and structural equations parameterised by $\psi$ were trained with a learning rate of $10^{-2}$ for a maximum number of 1000 epochs, batch size $B = 50$, and with the Adam optimiser [Kingma and Ba, 2014]. A total number of $R = 1000$ training samples was generated from the ABM for each of the observational and interventional training sets; these were each split 5 times into different training and validation sets of sizes 800 and 200, respectively, with a new surrogate model trained from scratch on each of these splits. We apply an early stopping criterion in which training is ceased if the validation error does not decrease for 20 consecutive epochs.

## E.7 Additional Case Study

In this case study, we consider a different policy scenario: reintroducing a species into an ecology, and simulating the ensuing population dynamics. Specifically, we adapt slightly a model from Wilensky and Reisman [2006]: we model an environment initially consisting of grass, sheep, and wolves, in which grass grows and is eaten by sheep, sheep eat grass and reproduce and are eaten by wolves, and wolves eat sheep and reproduce. The intervention we consider entails reintroducing a third animal species – bears, which eat both sheep and wolves, and also reproduce – whose population is originally zero but is made non-zero at some intervention time $t$. We imagine that $t$ is the variable the policymaker wants to optimise here.

We simulate the interactions between these four species in a spatial model, in which members of each animal species move around the grid and interact with the other species. We are then interested in understanding how the reintroduction of the bears affects the overall population dynamics, i.e., the counts of each animal in each species, along with the quantity of grass over time. As in the epidemic case study, we consider the problem of learning interventionally consistent surrogates for this complex spatiotemporal simulator, and once again examine three possible approaches for constructing surrogate families:

1. a family of deterministic mechanistic models based on a discrete-time Lotka-Volterra model of population dynamics [see, e.g., Sabo, 2005], where (analogously to the LODE surrogate family discussed in the epidemic case study) the underlying deterministic dynamics of the population dynamics model index a probability distribution at each time step (in this case, a Binomial distribution for each of the 4 species);

2. an LRNN family, exactly mirroring the LRNN family considered in the epidemic case study presented already;

3. and a third family considers a hybrid approach, where (as in the LODE-RNN family considered in the epidemic case study) we pass a recurrent network over the underlying Lotka-Volterra-type population dynamics model first before taking the output of the recurrent network to index the Binomial distributions for each of the four species.

A table for the results of this additional case study is shown in Table 2, where we see that the results are qualitatively very similar to the epidemic case study already presented: we see that training surrogates using our framework yields significant improvements in the surrogates' interventional consistency over observationally trained baselines, and that interventionally trained surrogates only see a minor decrease in performance on observational data compared to the drop in performance the observational surrogates see on interventional data.

Table 2: Metrics for interventionally ($\mathbf{I}$) & observationally ($\mathbf{O}$) trained surrogates on interventional ($\mathbf{I}'$) & observational ($\mathbf{O}'$) test sets (median$_{\text{first quartile}}^{\text{third quartile}}$ from 5 repeats) for the predator-prey case study. AMSE & ANLL measure ability to model counts of each species over time. **Bold** denotes best performance.

| Test | Model | LRNN | | LODE-RNN | | LODE | |
|------|-------|------|------|----------|------|------|------|
| | Train | **I** | **O** | **I** | **O** | **I** | **O** |
| $\mathbf{I}'$ | AMSE ($\times 10^2$) | $\mathbf{1.65_{1.62}^{1.80}}$ | $3.30_{2.37}^{4.32}$ | $\mathbf{1.79_{1.79}^{1.83}}$ | $2.23_{2.13}^{2.29}$ | $\mathbf{41.84_{40.13}^{44.74}}$ | $201.78_{46.11}^{250.80}$ |
| | ANLL ($\times 10^3$) | $\mathbf{0.78_{0.77}^{0.81}}$ | $11.20_{10.09}^{11.64}$ | $\mathbf{0.93_{0.92}^{0.94}}$ | $5.63_{4.72}^{5.96}$ | $\mathbf{6.54_{6.44}^{10.14}}$ | $47.14_{37.33}^{50.33}$ |
| $\mathbf{O}'$ | AMSE ($\times 10^2$) | $1.85_{1.81}^{2.03}$ | $\mathbf{1.61_{1.59}^{1.64}}$ | $2.18_{2.08}^{2.19}$ | $\mathbf{1.56_{1.51}^{1.67}}$ | $63.89_{38.25}^{326.33}$ | $\mathbf{36.75_{36.08}^{38.25}}$ |
| | ANLL ($\times 10^3$) | $0.76_{0.71}^{0.80}$ | $\mathbf{0.69_{0.68}^{0.70}}$ | $1.25_{1.24}^{1.25}$ | $\mathbf{0.66_{0.65}^{0.70}}$ | $32.11_{10.56}^{33.90}$ | $\mathbf{5.49_{4.96}^{6.18}}$ |

