# OpenReview forum: "Interventionally Consistent Surrogates for Complex Simulation Models"
_NeurIPS.cc/2024/Conference — NeurIPS 2024 poster_

### Official Review · Reviewer_3hmr · 2024-06-23

**Soundness:** 3
**Presentation:** 3
**Contribution:** 3
**Rating:** 6
**Confidence:** 3

**Summary:**

In their use by practitioners, complex simulators may be run under intervention scenarii. To reduce the overall cost of running many scenarii, relying on cheaper to compute surrogate models is a standard practice. The authors thus propose surrogates that can additionally learn these interventions effect, relying on a causal inference representation. Specifically, it divides the training into a parametric surrogate family and abstraction map parameters that are learned with the Kullback-Leibler divergence. An epidemiology example is provided to show the interest of learning this decomposition on surrogate consistency.

**Strengths:**

- the paper discuss a relevant practical use case for surrogate modeling
- the concepts are clearly described under the low page limit

**Weaknesses:**

- the discussion of the results could be improved
- the test case is perhaps a bit simple

**Questions:**

- the naive way for surrogate modeling in this context (say with neural networks or random forests) would be to also give as inputs the intervention parameters corresponding to the observation. It is not completely clear to me if this is the case for the RNN in the example.
- More analysis on the performance of interventionally trained data on observational data would be beneficial.
- perhaps some discussion on surrogates with causal inference would be relevant, see e.g.,
Witty, S., Takatsu, K., Jensen, D., & Mansinghka, V. (2020, November). Causal inference using Gaussian processes with structured latent confounders. In International Conference on Machine Learning (pp. 10313-10323). PMLR or Toth, C., Lorch, L., Knoll, C., Krause, A., Pernkopf, F., Peharz, R., & Von Kügelgen, J. (2022). Active bayesian causal inference. Advances in Neural Information Processing Systems, 35, 16261-16275.
- A related topic is the use of monotonicity information, e.g., on intervention effect, see e.g., Riihimäki, J., & Vehtari, A. (2010, March). Gaussian processes with monotonicity information. In Proceedings of the thirteenth international conference on artificial intelligence and statistics (pp. 645-652). JMLR Workshop and Conference Proceedings.

---

> ### Author Rebuttal · Authors · 2024-08-06
>
> **Improving the discussion of results**
>
> We will use some of the additional space in the revision to provide further discussion. In particular, we will:
> - state and discuss the relative computational costs of our complex simulators and our surrogates (briefly: we see that our surrogates run approximately 3-30 times faster across the experiments we consider); and
> - provide a more extensive discussion of the advantages of training surrogates for interventional consistency. For example, with respect to the epidemic case study we present, we can discuss a hypothetical scenario in which a policymaker must decide whether and when to introduce a lockdown in order to minimise the average number of infections over the simulated time horizon. Please see the Global Rebuttal for further details. (Is this the kind of discussion you would like to see? We hope this provides the discussion you are looking for on the relative performance of the interventional vs observational surrogates (as in your second question); please do let us know in the discussion period if there are specific points you would like us to discuss.)
>
> **Further experiments**
>
> To address your point that "the test case is perhaps a bit simple": the reason we focus on this particular epidemic simulator is that it allowed us to clearly and transparently demonstrate that simulators can be expressed as SCMs, which is important for establishing that the theory of causal abstraction can be used to learn surrogates. To nonetheless demonstrate our framework in a further, more complex setting, we have prepared another case study in which we model an ecology of species in a predator-prey relationship and the possible effects of reintroducing an additional species into the mix. Please see the Global Rebuttal for details. We emphasise that this additional case study is indeed more complex, as requested, in that it consists of agents that move around a spatial environment and reproduce and die, and consists of stocks of natural resources that grow and are consumed over time. We hope this addresses this comment, and that it provides further evidence that our framework can be applied in various settings.
>
> **Question about alternative approaches to surrogates and RNN**
>
> We believe the approach you suggest could in principle be taken for a limited set of surrogate families. (This is not what we do in the `LRNN` model: as described in the Appendix, we feed $\tilde{\boldsymbol{\theta}}$ into the surrogate at each time step, such that intervening at time $t$ amounts to feeding in a modified $\tilde{\boldsymbol{\theta}}$ at that time.)
>
> However, this will not work generally. For example, it would not work when the surrogate family is mechanistic, such as in the case of the `LODE` family we consider in our experiments. Thus it can't be the basis of a general framework for training interventionally consistent surrogates. In contrast, our framework is useful because it handles both cases, and is thus more general.
>
> This approach would also have the undesirable property that an intervention at time $t$ would have a causal influence on variables "located" at times $<t$, which by definition should not be the case. This would mean that time has different meanings in the two models, which amongst other things will reduce interpretability.
>
> Note that interventions in our surrogates are explicitly specified on abstract causal variables via $\omega$. Consider a pandemic sim where music venues can be closed. In our surrogates, $\omega$ specifies how venue closure can be interpreted as a direct intervention on, say, $\alpha$ in the `LODE` surrogate. If interventions aren't transformed via $\omega$ you lose this interpretability. You also lose out on scalability: $\omega$ reduces the space of possible interventions significantly.
>
> **Literature recommendations**
>
> Similarly to us [R4.1] & [R4.2] focus on improving causal reasoning methods in complex systems with causal inference. [R4.1] investigates hierarchical data settings with latent confounders, while [R4.2]'s active learning framework performs inference without assuming access to the true SCM. Instead, it assigns a Bayesian prior over the causal model class & jointly infers a posterior over both causal models (discovery) & queries of interest (inference). Both frameworks aim to handle complex causal relations which is crucial for surrogate modelling of complex systems.
>
> Additionally, both frameworks employ GPs to model causal relationships, aligning with our objective of building surrogates when it's hard to work directly with the underlying SCM. However, they rely on several simplifying structural assumptions in order to restrict their focus to a class of causal sufficient/identifiable models. In contrast, we allow for a wider class of SCMs, resulting in a more robust framework. Further, these papers overlook the interventional consistency between the surrogate & simulator, a topic we thoroughly examine via causal abstraction. A notable distinction is our approach for addressing the computational difficulties of large-scale simulators, especially within policymaking scenarios, through a causally consistent & computationally efficient surrogate paradigm. Our goal of developing interventionally consistent surrogates aligns with the goal of accurately estimating causal effects discussed in these papers. We will update the Related Work section of the revision, & include [R4.1] & [R4.2] as examples of surrogates for causal inference.
>
> [R4.3] is indeed relevant to our work when domain expertise is available & can be incorporated into the surrogate by enforcing monotonicity on certain features. As noted, an example could be when we identify a monotonic effect in interventions. For instance, one may enforce that longer lockdowns lead to less infections. We'll incorporate this discussion into the revision.
>
> **Refs**
>
> [R4.1] Witty et al. (2020)
>
> [R4.2] Toth et al. (2022)
>
> [R4.3] Riihimäki et al. (2010)

---

> > ### Comment · Reviewer_3hmr · 2024-08-14
> >
> > Thank you for the detailed replies to my comments and for proposing a newer test case. I think there is room for this work that relies partly on domain knowledge, but there is an effort to include the above responses.

---

> > > ### Author Response · Authors · 2024-08-14
> > >
> > > Thank you for your feedback and for your considered assessment of our work. If you believe our improvements warrant an increase in your initial score, we would greatly appreciate if this could be reflected in your updated score prior to the deadline in ~25 mins.

---

### Official Review · Reviewer_x5Gt · 2024-07-05

**Soundness:** 3
**Presentation:** 3
**Contribution:** 1
**Rating:** 2
**Confidence:** 4

**Summary:**

This work presents a method for learning surrogate models of simulations.
Both the simulation and the surrogate model are treated as causal models, where the surrogate is an element of a parametric family of causal models.
The map from simulation variables to surrogate variables is assumed to be known.
For training the surrogate, interventional consistency is used: interventions in the simulation should be well-represented by the surrogate.
That is, a map from simulation interventions to surrogate interventions has to be learned.
As a training loss, an abstraction error is used, which measures the distance between the pushforward of intervened simulation variables and intervened surrogate variables.
The theory section shows that zero approximation error leads to an exact transformation and that approximation errors for a given intervention are bounded.
The approach is demonstrated on an agent-based simulation of an epidemic model (SIRS), with interventions corresponding to different spreading rate settings (some represent lock-downs).

**Strengths:**

- The paper is a pleasure to read: the writing is well-structured and very clear.
- The mathematical formalism is well-defined, consistent and easy to follow.
- Thus far, the causal abstraction literature has mostly focussed on the theoretical foundations and mathematical structure of abstractions. This paper shows an interesting application of causal abstractions and thereby bridges the gap between the causality ivory tower and useful applications.

**Weaknesses:**

While I very much enjoyed reading the paper, it is unclear to me what its contribution is with respect to previous literature. The main elements (e.g. treating simulations as causal models and learning a simpler surrogate, making the connection to causal abstractions and using some form of approximation error for training) have been introduced before in [1] with only minor differences as far as I can tell.

References:

[1] Kekić et al. "Targeted Reduction of Causal Models." UAI (2024)

**Questions:**

- L96 "we restrict our attention to hard interventions": It looks like you don't use this fact later on in the theory. Could you relax this requirement? In fact, you could argue that the interventions applied in the case study look more like soft interventions. You intervene on $\theta$, which really just encodes the parameters of the causal mechanisms responsible for propagating infections and other state variables through time. So theta looks more like something that encodes some mechanisms, rather than a state variable.
- L216 "we assume that the base model M is implicitly represented by a simulation model of a complex socio-technical system": Is this just a helpful picture to have in mind for the reader, or is there some technical assumption that you make? It doesn't seem to me that this is used later on. Maybe I'm confused by "socio-technical system": for what type of simulations would the method not work?
- Fig. 4: Why are there edges from $\tilde{\theta}$ to $\tilde{y}_2$ and to $\tilde{y}_3$? I thought $\tilde{\theta}_t$ encodes the transition parameters from one time step to the next. Similarly, for $\tilde{I}_0$: wouldn't it only affect the first time step?

**Limitations:**

The method has some severe limitations in how easily it can be applied to real-world simulations, as it does not specify how to tackle the following:

- How to make the distribution of the surrogate family tractable and differentiable.
- How to select the surrogate family.
- How to find the variable map between the simulation and the surrogate.

These elements are assumed to be given, but I think they are the main difficulty when modelling simulations. I understand that modellers need to provide some level of domain expertise and that makes every application somewhat different. But the paper doesn't give much guidance on how the difficult points above can be addressed.

While Prop. 1 and 2 are reassuring, they don't give the modeller many insights as to how the approach behaves. For example, how do you need to choose the set of interventions in order to get a good surrogate? That is touched on in the case study, but to make the method more reliable in practice it would be great to have some more general theoretical analysis. Another open question is: What happens when you misspecify the surrogate by choosing a causal structure that doesn't match that of the simulation?

---

> ### Author Rebuttal · Authors · 2024-08-06
>
> We thank the reviewer for their time & kind comments ("paper is a pleasure to read..very clear", "well-defined, consistent and easy to follow", "bridges the gap between.. causality.. and useful applications").
>
> The reviewer identifies as a single weakness the novelty of our paper wrt a very recent UAI 2024 paper by Kekić et. al. [R3.1], presented at UAI less than 3 weeks ago. Despite this being categorized as a contemporaneous work by the NeurIPS Reviewer Guidelines, we are happy to detail the main differences between the two papers, highlight our novel contributions, & discuss this contemporaneous work in our paper.
>
> The two papers share some similarities as they both start from an existing causal abstraction framework, adapt a measure of interventional consistency, and use it learn new models. However the two works differ substantially in aims, methodology & results:
>
> - [R3.1] has an _explanatory focus_. They deal with _targeted reduction of causal models_ aimed to generate a model "explaining causal influences on an observable target variable $Y$"; they describe the target variable $Y$ as a "property of interest" & the abstraction to $Y$ as a "detector" quantifying "the presence or magnitude of a phenomenon in the data". Thus, reduction works around a chosen focal point ($Y$) & the only causal dynamics of interest are the ones converging on the said variable. In contrast, our work has a _simulation focus_: we deal with _learning surrogate models_ that would capture the whole causal dynamics of a system of interest at a different level of abstraction. Our work does not require the modeller to commit to any target variable: in the context of our SIRS simulation, [R3.1]'s approach would require the modeller to choose which target variable is of interest (e.g.: number of infected or number of recovered); our approach, instead, provides a simplified & accurate surrogate that simulates the dynamics of all relevant variables. To provide analogous simulation results, [R3.1]'s approach would require instantiating as many reduced models as the variables of interest
> - There are also fundamental methodological differences. Since reduction is interested in a single target variable, only constructive transformations equivalent to clustering are considered & the reduced causal model is a simple collection of nodes $Z_i$ only affecting node $Y$. This trivial structure does not encode complex causal dynamics, such as influences between variables $Z_i$; it has indeed a resemblance to ICA, as explained in their discussion. On the other hand, we consider a broader class of $\tau$ maps and arbitrary surrogate models describing complex causal structures
> - [R3.1] introduces further simplifications (linearity of $\tau, \omega$, affinity of the high-level mechanisms) for the sake of identifiability analysis. Experiments are run on simple physical models, as these simplifications rarely comply with real-world systems. Not requiring them, our approach is applied to actual ABMs which, as the reviewer states, "bridges the gap between the causality ivory tower and useful applications"
>
> The most striking similitude between our works is how we adapted the existing interventional consistency loss by replacing a JS distance with a KL divergence, & substituting a max operator with an expectation. But these are common simplifications in ML adopted in causal abstraction papers predating both our works. Interestingly, from a similar loss function, we derived propositions that are relevant wrt our different aim: [R3.1] proves positivity, invariance to invertible reparametrization & zero for exactness; we prove zero for exactness and upper-bound for divergence.
>
> We will briefly discuss this contemporaneous work in the Related Work section & add in the Appendix an exhaustive comparison along the lines above. A detailed comparison will be valuable to practitioners for choosing the approach that best suits their needs, whether modelling locally the dynamics of a single variable or whether simulating an entire system at a coarser scale.
>
> **Questions & limitations**
>
> Q1: Hard interventions are a requirement of the used abstraction framework as the loss is defined between posets of interventions which arise only in such a case. Our interventions are defined on parameters controlling the dynamics of the system; this modelling choice allows us to: disentangle the dynamics of the system & the params controlling it; rely on known ODEs for the dynamics; express parsimonious (defined by a single value) & interpretable (lockdown equal to setting the param to zero) interventions. In general, our work may be expanded to deal with soft interventions [R3.2], e.g., to model realistic uncertain/"fat-fingers" interventions by policy-makers.
>
> Q2: Just a helpful picture! The method is general.
>
> Q3: The `\tilde{\theta}_t` determine the update of the hidden state `z_t` of the ODE/RNN at time `t`, and `z_t` mediates the dependency of `\tilde{y}_t` on the `\tilde{\theta}_{1:t}` and `I_0` (please see Appendix E). However, we have wrongly put arrows from `\tilde{y}_t` to `\tilde{y}_{t+1}`; will remove in revision.
>
> L: It is incorrect that we do not specify "how to make the distribution of the surrogate family tractable and differentiable": we show three examples for how this can be done by combining neural networks, differentiable simulators, & standard distributions in the experimental section. We will nonetheless discuss other strategies in the revision, e.g. the use of neural generative models (e.g., normalising flows) to specify the stochasticity of surrogates, & of sample-based interventional consistency losses (e.g., MMDs) for the case of differentiable surrogates with intractable distributions.
>
> For other limitations, please see our answer to Reviewer fR4E: specifically, on choosing interventions/finding a variable map/misspecification see Q1/L1/Q2.
>
> **Refs**
>
> [R3.2] Massidda et al. "Causal Abstraction with Soft Interventions." CLeaR (2022)

---

> > ### Comment · Reviewer_x5Gt · 2024-08-08
> >
> > Thank you for providing a detailed rebuttal to my review. I went over the rebuttal, the other reviews and had another look at the paper and will try to comment on the rebuttal below.
> >
> > ## On the contributions with respect to prior work
> >
> > From the section on contemporaneous work from the [Call for Papers](https://neurips.cc/Conferences/2024/CallForPapers):
> > > For the purpose of the reviewing process, papers that appeared online within two months of a submission will generally be considered "contemporaneous" in the sense that the submission will not be rejected on the basis of the comparison to contemporaneous work. Authors are still expected to cite and discuss contemporaneous work and perform empirical comparisons to the degree feasible.
> >
> > Since [R3.1] appeared online in November 2023 (see the [arxiv version](https://arxiv.org/abs/2311.18639)) it is not considered contemporaneous work for this submission, as the submission deadline was in May 2024.
> >
> > The paper's contributions, as summarised in the first part of the paragraph starting in L 49:
> > > To address this, we build on recent developments in causal abstraction [...]. We view the complex simulator and its surrogate as structural causal models [...], and propose a framework for constructing and learning surrogate models for expensive simulators of complex socio-technical systems that are interventionally consistent, in the sense that they (approximately) preserve the behaviour of the simulator under equivalent policy interventions. This perspective enables treating the surrogate model as a causal abstraction of the simulator.
> >
> > The points in the part above have been covered in [R3.1] (with some minor differences that were mentionned in the rebuttal and which we will discuss below). Such that the statement in L58:
> > >Our approach establishes, for the first time, a connection between complex simulation models and causal abstraction, and a practical approach to learning interventionally consistent surrogates for complex simulators.
> >
> > is incorrect (same goes for L369).
> >
> > Now we can consider the merits of the paper regarding the second part of the contributions (L55):
> > > We motivate our proposed methodology theoretically, and demonstrate with simulation studies that our method permits us to learn an abstracted surrogate model for an epidemiological agent-based model that behaves consistently in multiple interventional regimes.
> >
> > and as an extension of prior work.
> >
> > The epidemiological case study in Sec. 5, does encode more complex causal structure and has a nonlinear (hand-crafted, rather than learned) variable map from the simulation to the surrogate.
> > However, as discussed in the limitations section of my review, the method does not specify how to find them.
> > Prior work [R3.1] defines a method to learn the maps between variables and interventions under some simplifications (as outlined in the rebuttal).
> > This submission, however, generalises this to exactly one case study (or two if you count the results promised in the rebuttal) and does not provide a method that is directly applicable to the wider ML/simulation community.
> > The crucial parts are assumed to be given through domain expertise and/or not learned by the method.
> > The part that is general about the proposed approach (Sec. 4) vague and most of the heavy lifting has to be done by the person that runs the simulation.
> >
> > Furthermore, in order to consider this an extension of prior work you would expect an experimental comparison which is also missing. The comparison could be done by "instantiating as many reduced models as the variables of interest" as mentionned in the rebuttal or choosing one variable to compare against.
> >
> > I commented above on the theoretical results in the initial review.
> >
> > Therefore, besides some of the main contributions having been covered by prior work, I think the limitations outweigh the contributions of the proposed approach when viewed as an extension of it.
> >
> > ## Questions
> >
> > Q1: Couldn't you always consider the parameters defining the causal mechanism as a separate variable and intervene on this separate parameter variable? Suppose your soft intervention changes the mechanism from the original one to a member of a family of mechanisms and the parameter variable acts as a "mechanism selector". In that case, wouldn't the approach be applicable?
> >
> > ## Conclusion
> >
> > I appreciate the effort put into the rebuttal and the clarifications provided. However, the primary concerns regarding the novelty and applicability of the proposed method, especially in light of prior work, remain.

---

> > > ### Author Response · Authors · 2024-08-09
> > > **(1/2)**
> > >
> > > (1/2)
> > >
> > > Thanks for your response. We provide a summary TL;DR before providing a detailed response in our next comment.
> > >
> > > ## TL;DR
> > >
> > > 1) **[R3.1] is entirely unsuitable** for our setting:
> > >
> > >     a) **[R3.1] does not handle multiple interdependent target variables**;
> > >
> > >     b) **[R3.1] does not handle discrete or bounded target variables**; and
> > >
> > >     c) **[R3.1] scales extremely poorly** requiring $\sim 10^7$ trainable parameters for our epidemic case study, and _billions_ of parameters for more realistic simulators.
> > > In contrast, **our approach does not suffer these limitations**.
> > >
> > > 2) We disagree that fixing $\tau$ does the "heavy lifting": **knowing the concepts you care about doesn't mean you know the mechanism that relates them**. Moreover, we believe that fixing an interpretable $\tau$ is easier than setting opaque hyperparameters to learn a potentially uninterpretable $\tau$, as in [R3.1].
> > >
> > > 3) **Methods for nonlinear and non-Gaussian abstractions are recognised by [R3.1] _themselves_ as important**. Our work has developed such a method contemporaneously with [R3.1].
> > >
> > > Whilst we believe our work is contemporaneous with [R3.1], we are still consulting the ACs about the evidence we can provide without breaching review policy. Thus, we will address the point regarding arXiv in a later comment.

---

> > > > ### Author Response · Authors · 2024-08-09
> > > > **(2/2)**
> > > >
> > > > (2/2)
> > > >
> > > > ## Unsuitability of [R3.1]
> > > >
> > > > > ...you would expect an experimental comparison which is also missing.
> > > >
> > > > Suppose we attempt an experimental comparison by applying [R3.1]'s method (herein TCR) to our epidemic case study. The reviewer agrees that we should do so by "instantiating as many reduced models as the variables of interest". In our epidemic case study, there are $3 T$ such variables – each of the $S_t, I_t,$ and $R_t$.
> > > >
> > > > **TCR's first limitation**: the population size in our model is fixed to $N$, such that $S_t + I_t + R_t = N$. This cannot be enforced in TCR, since TCR models $S_t, I_t,$ and $R_t$ separately (Sec. 4 of [R3.1]: "A target scalar variable $Y$...").
> > > >
> > > > **TCR's second limitation**: $S_t, I_t,$ and $R_t$ are all integer-valued and bounded in $[0, N]$. This is not handled by TCR, since TCR assumes target variables are Gaussian (p.6 of [R3.1]: "...we make a Gaussian assumption on the densities...").
> > > >
> > > > Suppose we proceed with TCR despite these limitations. Following [R3.1]'s experiments, a TCR model at time $t$ would consist of $\tau$ and $\omega$ maps from the $Nt$ states of the agents over time to a target variable at time $t+1$. ([R3.1]'s double well experiment targets the final position of a mass in a double well by learning a TCR from the positions & velocities of the mass at all preceding times. See e.g. Line 115, `base.py` in their GitHub repo.) **TCR's third limitation**: For high-level models with $n$ causes, the number of free parameters across all TCR models would be of order $\sum_{t=1}^T NtnD \sim NT^2Dn$, where $D$ is the number of target variables per time step. For the epidemic case study we have presented ($D=3$, $N=2500, T=50$),  **[R3.1]'s approach would require ~$\mathbf{10^7}$ free parameters** compared to the $\sim 10^3-10^4$ free parameters used in our surrogates, even when only one high level cause is adopted ($n=1$). For larger-scale models, [R3.1]'s approach would be entirely inapplicable: to find a targeted causal reduction for Covid spread amongst a few million agents over a few months, **[R3.1]'s approach would require billions of free parameters**.
> > > >
> > > > ## Fixing $\tau$
> > > >
> > > > > The epidemiological case study... has a nonlinear (hand-crafted, rather than learned) variable map... However... the method does not specify how to find them.
> > > >
> > > > > ...crucial parts are assumed to be given through domain expertise... most of the heavy lifting has to be done by the person that runs the simulation.
> > > >
> > > > We disagree that setting $\tau$ is where most *heavy lifting* occurs. Just because a set of variables have been identified via $\tau$, it does not mean the domain expert is aware of how these variables affect and interact with each other. For example, one may be aware of temperature and pressure in a physical system, but unaware of how they relate to each other. That is, **much of the heavy lifting lies in identifying the underlying mechanism** between emergent system properties. As already stated, [R3.1] avoids much of this issue by assuming linearity and Gaussianity, whilst we do not. **[R3.1] themselves** highlight these assumptions as problematic when modelling large scale complex systems:
> > > >
> > > > >  ...we made Gaussian approximations and used linear 𝝉 and 𝝎 maps. While this has clear benefits, this may be too limiting for some complex simulations...
> > > >
> > > > Also, the experiments of [R3.1] rely on expert knowledge, just in a different way. In each experiment the appropriate number of causal variables are chosen to model the phenomena of interest:
> > > >
> > > > - Double well: 1 variable capturing the mass' momentum at the well's apex.
> > > >
> > > > - Mass Spring: 2 variables representing independent shifts in the $x$ and $y$ directions.
> > > >
> > > > Choosing the right number of causal variables is challenging for large-scale complex systems, and it is unclear how one would do so without knowledge of macroscopic properties. Moreover, [R3.1] employs a regulariser to ensure that all causal variables are used by $\tau$ and $\omega$. Thus, if too many variables are specified, TCR may learn a less effective and interpretable map. We have no such issue. Lastly, note that using a $\tau$ informed by experts ensures the surrogate is **easy to interpret** in terms of macroscopic system properties. This isn't guaranteed when $\tau$ is learned.
> > > >
> > > > (To be clear: we assume $\tau$ is given, but we learn $\omega$.)
> > > >
> > > > ## Generality of Approach
> > > >
> > > > One only requires a $\tau$ map and a surrogate family to run our approach. Moreover, the surrogate families used in our SIR experiments are very generic. This is evidenced by our ecological example, which relies on the same families. In general, it's easy to specify a surrogate family that meets our assumptions. Many complex systems are studied via ODEs/system dynamics models that can be used as a basis. (This is how some of our classes of surrogates are motivated.)

---

> > > > > ### Author Response · Authors · 2024-08-13
> > > > >
> > > > > As further evidence of the inapplicability of [R3.1]'s work to the settings we consider (and, therefore, as further evidence of the significance of our contribution), we have run an experimental comparison against [R3.1]'s approach. We used their code on GitHub (released in June, a month after the NeurIPS submission deadline) and followed the setup for their "double well" experiment. Details are below.
> > > > >
> > > > > **Key takeaways**:
> > > > >
> > > > > - To run [R3.1]'s approach, we were forced to run reduced experiments on a far smaller scale (`N=64` agents, `T=20` time steps) compared to what we are able to do with our own methods, because of the extremely poor scalability of [R3.1]'s approach (which we highlighted in our previous response).
> > > > > - Even in this extremely small-scale example, [R3.1]'s approach demands ~150k trainable parameters, and requires ~6 hours of training to learn high-level models for all `3T` `S_t`, `I_t`, and `R_t` variables. On this basis alone, it does not compete with our methods: our (off-the-shelf) `LRNN` surrogate family requires ~2 mins to train a single high-level surrogate simulator for _all_ `S_t`, `I_t`, and `R_t` variables.
> > > > > - Even in this extremely small-scale example, [R3.1]'s approach achieves an MSE loss of ~8.7 on its ability to recover the `S_t` alone. In contrast, our `LRNN` surrogate family achieves an MSE loss of ~0.6 _across all_ `S_t`, `I_t`, and `R_t`. The `LRNN` achieves this by learning (a) how to map interventions from the base model to the abstract model (through a learnable $\omega_{\phi}$) and (b) the mechanisms governing the evolution of the high-level variables (i.e. through learnable surrogate parameters $\psi$).
> > > > >
> > > > > For these reasons, regardless of whether [R3.1] constitutes prior or contemporaneous work, our contribution is significant and impactful: currently, our approach provides the only viable (practical, scalable, and interpretable) method to learn interventionally consistent surrogates for complex simulators of the kind that are used to inform decision-making in complex systems.
> > > > >
> > > > > In light of this and of the discussion above, the reviewer's score and position – which is based on their "primary concerns regarding the novelty and applicability of the proposed method, especially in light of prior work" – is not justified.
> > > > >
> > > > > ### Details
> > > > >
> > > > > In [R3.1]'s double well experiment, the motion of a ball is simulated according to an equation of motion corresponding to a double well potential. The ball's velocity is shifted at different time steps, while the ball's position is unshifted throughout (see App. F.2 of their paper). The authors of [R3.1] learn $\tau$ and $\omega$ maps from the ball's positions and velocities over all times $t=1,...,T-1$ to the final position of the ball at end time $T$ (see Sec. 5.2 of their paper). Following this example, learning a targeted causal reduction to a variable at time $t$ would involve learning $\tau$ and $\omega$ maps from all variables in the low-level graph corresponding to times preceding $t$. Therefore, applying [R3.1]'s approach in our setting involves learning $\tau$ and $\omega$ maps from all agent states (the $N$-dimensional $x_t$ in our notation) and parameters ($\theta_t$ in our notation) for $t=1,...,t'-1$ for any target variable occurring at time $t'$. We train both [R3.1]'s approach and our `LRNN` surrogate for 40 epochs ([R3.1]'s approach is too computationally demanding to train for longer) and train for 5 different repeats.

---

> > > > > > ### Comment · Reviewer_x5Gt · 2024-08-14
> > > > > >
> > > > > > Thank you for providing additional clarifications and arguments. To me, it seems we are converging to a state where a big chunk of the arguments are repeated, so I'll try to be brief in my answer.
> > > > > >
> > > > > > ## On the comparison to [R3.1]
> > > > > >
> > > > > > Of course, a handcrafted solution to a specific problem scales better (in terms of compute and number of parameters) and achieves better losses than a general-purpose method.
> > > > > > If I want to fit a function and I know it's a polynomial, I can achieve better scalability and results by using a polynomial inductive bias rather than, say, a neural network.
> > > > > >
> > > > > > I think the surrogate learned for the epidemiological case study is appropriate and summarizes the original system well.
> > > > > > But besides the submission claiming false novelty, my main criticism is that it doesn't provide a generally applicable method.
> > > > > >
> > > > > > ## Summary
> > > > > >
> > > > > > While I appreciate the effort of adding detailed answers in the discussion, I'm afraid the main criticisms haven't been resolved:
> > > > > >
> > > > > > - The paper makes inaccurate claims of novelty, since a large part of the reported contributions have been covered by prior work.
> > > > > > - The paper presents a case study for learning surrogates for one application (or two if you take the example promised during the rebuttal). However, it relies on domain knowledge to define crucial elements of the surrogate map and model. Therefore, it is of limited use to the wider ML/simulation community.
> > > > > >
> > > > > > Therefore, I will keep the score of 2 based on "limited impact" (as in the definition of the score).

---

> > > > > > > ### Author Response · Authors · 2024-08-14
> > > > > > >
> > > > > > > - Reviewer `x5Gt` continues to call our solution "handcrafted", which _completely ignores_ the fact that we learn two elements that are crucial for the accuracy of the high-level simulator:
> > > > > > >
> > > > > > >     **a.** the omega map (which is crucial for correctly mapping interventions between models) and
> > > > > > >
> > > > > > >     **b.** the mechanisms governing the relationships between the high-level variables (which is crucial for simulating accurately from the high-level interventional distribution).
> > > > > > >
> > > > > > >     Therefore, Reviewer `x5Gt`'s claim that our solution is handcrafted is false, and appears to be a deliberate attempt to misrepresent our contribution.
> > > > > > > - Reviewer `x5Gt` claims that our contributions are not novel relative to [R3.1], yet we have demonstrated experimentally that the algorithm proposed by [R3.1] does not scale to our settings. In contrast, our own approach does. This drastic improvement cannot be achieved without novelty (else [R3.1]'s approach would already achieve this improvement). [R3.1] themselves highlight in their Limitations and Future Work section that "future work should explore more flexible approaches" than linear Gaussian abstractions, which further supports the novelty of our approach.
> > > > > > > - Reviewer `x5Gt` claims that our work is of limited use to the ML community, yet we run our learned surrogates on case studies that are _significantly larger_ in scale and complexity than those that [R3.1]'s approach can handle.
> > > > > > > - Reviewer `x5Gt`'s point about inductive biases actually highlights a strength of our method: we rely on _generic neural networks_ to learn the crucial components of our solution, while [R3.1] adopts a specific inductive bias of _linear_ Gaussian methods (even simpler than the tailored _polynomial_ method that Reviewer `x5Gt` offers themselves).
> > > > > > > - Reviewer `x5Gt`'s claim that our work is of "limited impact" is _incompatible with Reviewer `x5Gt`'s own assessment_ that our work "bridges the gap between the causality ivory tower and useful applications."
> > > > > > > - We have extensively covered the "problem" of determining $\tau$ in our discussion. Please see, for example, responses (1/2) and (2/2) to Reviewer `fUGP`. The domain knowledge we assume is only that the modeller knows what they are trying to model (which is always the case when a modeller builds a model), and is therefore minimal.
> > > > > > > - The additional case study is not "promised", but is expressly included in the Global Rebuttal and accompanying PDF file.

---

### Official Review · Reviewer_fUGP · 2024-07-05

**Soundness:** 3
**Presentation:** 3
**Contribution:** 3
**Rating:** 6
**Confidence:** 2

**Summary:**

The paper proposes a novel framework for learning surrogate models of expensive simulators by formulating them as structural causal models. Specifically, the authors focus on learning interventionally consistent surrogate models for large-scale complex simulation models. The authors' claims are supported by theoretical and empirical results.

**Strengths:**

a. The paper is well-written and easy to follow. Concepts are clearly explained and equations well-commented

b. Empirical results are sounding

**Weaknesses:**

a. Empirical tests seem to be limited in assessing the framework's true power. Tests on different, perhaps more complex, scenarios would help practitioners to better understand the benefits of the proposed framework and use it.

b.  One motivation for the proposed framework is to provide policy-makers with "fast" and high-fidelity surrogates to test the effect of different interventions. However, the empirical results do not clearly indicate what the advantage of surrogates compared to simulators is in terms of speed of simulations.

**Questions:**

See Weaknesses

**Limitations:**

Limitations of the work are reported in section 7

---

> ### Author Rebuttal · Authors · 2024-08-06
>
> Thank you for your review and suggestions for improving our paper. We provide itemised responses to your two questions below.
>
> **a.** To expand our empirical assessment of the framework we propose, and to provide practitioners with a further example of how they may use our framework, we have prepared a further case study that we will include in the revised paper. We provide details on this experiment in the Global Rebuttal above. We will also include a discussion of how causal surrogates can be benchmarked going forward, such as by assessing the degree to which the surrogates preserve the optimality or ordering of interventions in the original simulator with respect to some downstream optimisation task. Please see the Global Rebuttal for further details. We believe this would also be a useful guide to practitioners, and hope that this also addresses the reviewer's question here.
>
> **b.** We will include in the revision a comparison of the runtimes of the simulators and surrogates. Briefly: in the epidemic case study presented, the `LRNN`/`LODERNN`/`LODE` surrogates each run approx. 3 times faster than the ABM; in the new experiment described in the previous bullet point and in the Global Rebuttal, the surrogates run 10-30 times faster. Obviously, the exact speed up will be different for different simulators. That said, there are additional considerations that motivate the use of smaller-scale surrogates with tractable likelihood functions, such as interpretability, improved communication (e.g., in our epidemic case study, we would be able to explain to a decision-maker who is familiar with the `LODE` surrogate but not with the complex ABM that "applying intervention X in the ABM is like applying intervention Y in the `LODE`"), and the ability to optimise probabilities/probability densities directly, thus even potentially obviating the need for any simulation from the surrogate in any case. We will discuss this more fully, using some of the additional space in the revision, and the appendix if necessary.

---

> ### Comment · Reviewer_fUGP · 2024-08-12
>
> Dear authors, thank you for the rebuttal and the new experiments you mentioned in the global rebuttal.
>
> I still think that, although the topic is relevant for the ML community, there are strong limitations in its applicability by practitioners without strong domain expertise.

---

> > ### Author Response · Authors · 2024-08-12
> > **(1/2)**
> >
> > Thank you for your response.
> >
> > We are a bit unsure about what your comment about domain expertise is referring to without any additional context. We have nonetheless taken an educated guess as to your concern and address this below (TL;DR now; detailed response to follow).
> >
> > ### TL;DR
> >
> > 1) **Our reliance on domain knowledge is minimal** relative to some areas of interest to the NeurIPS community. Our assumption of a fixed $\tau$ map is entirely reasonable given that **modellers always know why they are building a model**. (Why would they build a model if they _didn't_ know this?) We will in any case include several detailed examples for how to choose $\tau$ in different settings; we include some of these examples below. (We believe this would also help to address the limitation listed by Reviewer `fR4E`.)
> >
> > 2) The principles of **keeping humans in the loop**, of **incorporating human knowledge into ML pipelines**, and **adequately evaluating a model's objectives prior to deployment**, often appear in guidelines on the **responsible use of AI** in decision-making (e.g., Principles 1, 3 & 6 of `The Institute for Ethical AI & Machine Learning`'s "Responsible Machine Learning Principles"; "Assessment and Accountability" in `The Center for AI and Digital Policy`'s "Universal Guidelines for AI"). Our assumption that there exists a domain expert in the loop who informs $\tau$ is therefore **consistent with the responsible use and development of interventionally consistent surrogates** and in this sense is a strength of our approach.

---

> > > ### Author Response · Authors · 2024-08-12
> > > **(2/2)**
> > >
> > > ## More detailed response
> > >
> > > 1) Using human knowledge & expertise is ubiquitous in areas of interest to the NeurIPS community. For instance, universal differential equations [1] require domain experts to specify an ODE roughly capturing the physical laws underlying a system. Further, many modern neural architectures rely on physical symmetries found through human domain expertise to improve efficiency (e.g.: permutation symmetries in graphs [2], equivariances in NNs for point cloud data [3], time reparameterisation invariance in NNs for time series [4]). Other active areas of research that incorporate strong biases from human knowledge include physics-informed NNs [5] & human-in-the-loop RL [6].
> > >
> > >    **Our use of domain knowledge is minimal** in comparison to these areas. We only require the modeller to supply the variables of interest to them so that $\tau$ is defined. (The modeller generally _will_ know what they are interested in using the model for: if they have sufficient domain knowledge to build a model, they will know what they want to achieve with the model, i.e. how to define $\tau$.) Our minimal reliance on human input is exemplified by our `LRNN` surrogate, which is an off-the-shelf/general purpose network that reproduces the complex simulators' behaviours of interest by taking in interventions through the learned $\omega$ map.
> > >
> > >    We will however provide in our revision several detailed practical examples (beyond the two case studies we present) illustrating how $\tau$ may be chosen for large-scale simulators, as a guideline for practitioners. Some examples we will expand on in the revision:
> > > - Consider the model of forced migration in [7]. Variables of interest to these modellers are the total number of displaced people by location over time by age, gender, and other demographic characteristics. $\tau$ would therefore be defined by counting the number of agents in each of these states at each location, i.e. $\tau_{l,d}(x_t) = \sum_{a\in A} \mathbb{I}[\text{agent }a\text{ has demographic features }d\text{ and is in location }l\text{ at time }t]$ where $l$ labels locations, $d$ are demographic features, $x_t$ is the state of the simulation at time $t$, $A$ is the set of all agents, & $\mathbb{I}$ is the indicator function.
> > > - Consider the model of flood risk mitigation behaviours in [8]. The modellers want to model what precautions households take to protect themselves from floods in high-flood-risk areas under different policy interventions. Households can: do nothing; purchase insurance; purchase property-level protection; or purchase property-level protection & insurance (see Fig. 3). Here, $\tau$ would count the number of households taking such actions in this case (as in the example above).
> > > - Consider the UK housing market model in [9]. Tables 2-6 define macroeconomic indicators such as inflation rates, unemployment rates & real interest rates that the modellers care about. Here, $\tau$ would be defined by standard macroeconomic formulae for these quantities.
> > >
> > > 2) Various guidelines on the **responsible use of ML** (e.g. those highlighted in our TL;DR) recommend that ML **should always be applied in conjunction with domain experts** to adhere to principles governing the responsible use of ML. This is especially true when it comes to high-stakes decision-making in complex systems (the setting we consider). In this way, rather than being a limitation of our approach, our assumption that $\tau$ is informed by a human expert provides a useful & important way to integrate human knowledge into the surrogate by allowing the decision-maker to fix the properties they care about for their decision-making – all while still allowing for flexibility through the learnable $\phi$ and $\psi$ parameters of the $\omega$ map & surrogate itself, respectively – & thus aligns with recommended guidelines for the responsible use of AI.
> > >
> > > [1] White et al. "Stabilized neural differential equations for learning dynamics with explicit constraints." NeurIPS (2023)
> > >
> > > [2] Gilmer et al. "Neural message passing for quantum chemistry" ICML (2017)
> > >
> > > [3] Schütt et al. "Equivariant message passing for the prediction of tensorial properties and molecular spectra." ICML (2021)
> > >
> > > [4] Kidger et al. "Deep signature transforms" NeurIPS (2019)
> > >
> > > [5] Krishnapriyan et al. "Characterizing possible failure modes in physics-informed neural networks." NeurIPS (2021)
> > >
> > > [6] Guan et al. "Widening the pipeline in human-guided reinforcement learning with explanation and context-aware data augmentation." NeurIPS (2021)
> > >
> > > [7] Ghorbani et al. "Flee 3: Flexible agent-based simulation for forced migration." Journal of Computational Science 81 (2024)
> > >
> > > [8] Geaves et al. "Integrating irrational behavior into flood risk models to test the outcomes of policy interventions." Risk Analysis (2024)
> > >
> > > [9] Bardoscia et al. "The impact of prudential regulations on the UK housing market and economy: insights from an agent-based model" Bank of England Working Paper (2024)

---

> ### Author Response · Authors · 2024-08-14
>
> Thank you again for your feedback – we hope we have been able to address your remaining concern about the limited role that domain knowledge plays in defining the variables of interest to the modeller. If you believe our improvements warrant an increase in your initial score, we would greatly appreciate seeing this be reflected in your updated score prior to the deadline in ~20 mins. Thank you again!

---

### Official Review · Reviewer_fR4E · 2024-07-15

**Soundness:** 3
**Presentation:** 3
**Contribution:** 3
**Rating:** 7
**Confidence:** 3

**Summary:**

The paper introduces a framework for learning surrogate models for complex simulations that preserve simulation behavior under changes in the structural parameters of the underlying model (i.e. the intervention) using causal inference. The framework is tested on epidemiological agent-based models of disease spread, against a set of ablative baselines.

**Strengths:**

S1: The paper addresses an important problem in the simulation of complex socio-technical, highly dynamical, systems; i.e. reducing the computational cost of running large-scale simulations when critical parameters of the model change. The proposed framework not only has potential to significantly accelerate experimentation in policy-adjacent fields, but also in other technical environments where that's the case (physics, many engineering sciences, etc).

S2: The theoretical foundation is solid, building on established concepts from causal inference and abstraction. The authors provide formal definitions and proofs for their key results on abstraction error and interventional consistency, and I found the reading relatively clear (even though causal models are not my field of work).

S3: The empirical evaluation on the SIRS epidemiological model provides a concrete demonstration of the method's effectiveness on what I think is a fairly representative model of the field (modulo again my superficial familiarity with such fields). The comparison between interventionally and observationally trained surrogates highlights the importance of considering interventional consistency, which neatly justifies both the paper narrative as well as the methodology.

S4: Overall the connection between complex simulation models and causal abstraction feels underexplored, and this work opens up new avenues for applying such techniques across the board.

**Weaknesses:**

W1: While the SIRS model provides a good initial test case, the empirical evaluation is limited to a single domain. It would strengthen the paper to demonstrate the method's applicability to other types of complex simulations, such as economic or social systems models.

W2: The paper does not provide a thorough comparison to existing surrogate modeling techniques beyond a basic observational training baseline. It would be extremely valuable to see how the proposed method compares to state-of-the-art surrogate modeling approaches, even if they don't explicitly consider interventional consistency.

W3: The scalability of the proposed method to very large and complex simulations is not thoroughly addressed. It's unclear how well the approach would work for simulations with hundreds or thousands of variables and complex interdependencies.

W4: The paper lacks a discussion of how to choose appropriate interventions for training the surrogate. In practice, the space of possible interventions may be very large, and it's not clear how to select a representative set for training and testing.

**Questions:**

Q1: How sensitive is the method to the choice of the interventional distribution $\eta$? How would you recommend practitioners choose this distribution in real-world applications? If you were to read the paper afresh, what would you need to have in the manuscript to be able to expand its experimental results so as to form a consistent set of benchmarks for the community?

Q2: Have you explored the performance of the method under model misspecification, i.e. where the surrogate family does not include a model that can perfectly capture the behavior of the original simulator?

Q3: The paper mentions that the method does not require explicit knowledge of the simulator's SCM. How does this compare to methods that do utilize such knowledge, and are there cases where having this knowledge would be beneficial (or where it's a necessity and/or the proposed method critically fails)?

Q4: How does the computational cost of training the surrogate compare to running the original simulator? Is there a break-even point in terms of the number of interventions one needs to evaluate for the surrogate to be worthwhile?

**Limitations:**

L1: The current framework assumes that the map $\tau$, which defines the aggregate quantities of interest, is pre-specified. In practice, determining the right level of abstraction and which quantities to preserve may be challenging. The paper would benefit from a discussion of how to choose appropriate τ maps.

---

> ### Author Rebuttal · Authors · 2024-08-06
>
> Q1 & W4: Generally, we expect $\eta$ to be informed by domain experts/policymakers based on downstream tasks, perhaps accounting for economic/political constraints. E.g. in a pandemic, economic constraints may preclude lockdowns of length >2 weeks, and political pressure may demand action soon; here, $\eta$ could be uniform over 2-week lockdowns starting within a month.
>
> We now turn to the reviewer's question on benchmarks. Whilst numerous datasets & benchmarks have been proposed in the causality literature [R1.1-3], they are typically confined to SCMs much smaller than those of large-scale simulators. Likewise, to our best knowledge, there is no consensus in the social simulation community regarding benchmark simulators, besides simple examples (e.g. the Schelling model). Further, benchmarking surrogates requires access to a set of realistic downstream tasks. For instance, to properly assess a pandemic surrogate one must know the problems epidemiologists are interested in. Summarising, benchmarking surrogate models of complex systems requires:
> - A set of large-scale standardised simulators spanning application domains.
> - A set of downstream tasks for each simulator on which surrogate models can be deployed.
>
> While we focus on presenting a new methodology for surrogate modelling grounded in causal abstraction theory, as we will discuss in revision, one way to devise a benchmark on downstream tasks is to check that interventional properties are preserved. For example, one can check that the internal ordering of interventions wrt their efficacy is preserved (see the Global Rebuttal).
>
> Q2: As recognised by the reviewer, the surrogate family may not fully capture the behaviour of the simulator. Note that the task of learning a surrogate may be formulated as minimising $\text{KL}(P \Vert Q(\psi, \phi))$ over $\Psi \times \Phi$. Here, $P$ is a joint distribution over interventions and abstract states given by first sampling an intervention $\iota \sim \eta$ and then sampling from the base model under $\iota$, while $Q(\psi, \phi)$ is a joint distribution defined by first sampling $\iota \sim \eta$ and then sampling from the corresponding interventional distribution in the surrogate. Given the form of this problem, classical results for maximum likelihood estimation can be adapted to provide misspecification guarantees; by leveraging [R1.4], one can show that our surrogate estimation is asymptotically normal w/ sandwich covariance & mean corresponding to the surrogate model w/ minimum KL-divergence to the perfect abstract model under $\tau$.
>
> Q3: Our method assumes no knowledge of the simulator's SCM. This ensures our method is generally applicable as it's difficult to directly characterise the SCM of large-scale simulators. It's possible that access to the base SCM/DAG may expedite abstraction by allowing us to focus on minimal intervention sets [R1.5-6], or leverage the identifiability of interventional distributions to reduce the number of simulations required from the base model [R1.7-8]. However, it's unclear that applying the do-calculus on large causal graphs is more efficient than simulating interventions directly. We will discuss these ideas in more detail in the appendices of the paper.
>
> Q4: Many of the surrogates we consider are neural networks, thus the time complexity of sampling an interventional outcome scales linearly w/ the dimension of interventions. Moreover, our neural surrogates benefit from parallelisation within each time step, massively reducing their run-time. In contrast, social simulators often rely on pairwise interactions between potentially millions of agents that are difficult to parallelise. Thus, once trained, neural surrogates can be orders of magnitude faster to evaluate than the simulator. Precisely characterising this speed up is non-trivial & depends on the internal structure of the simulator and the surrogate. For a comparison of runtime costs in our experiments, see part (b) of our response to Reviewer fUGP.
>
> Of course, there is a training cost associated with the surrogate, but this will be amortised over downstream tasks. That is, when the collective sample complexity of downstream tasks exceeds the sample complexity of causal abstraction, learning a surrogate is beneficial.
>
> Furthermore, running complex simulators may require technical expertise & high performance hardware not widely available. Providing interpretable surrogates that run on commodity hardware reduces the barrier to entry in such cases. Note that this mirrors the release of low memory LLMs that run on commodity hardware commonplace today.
>
> W1: Please see the Global Rebuttal.
>
> W2: Our baselines are already complex wrt SOTA in social simulation (see e.g. [R1.9]). Please let us know if there are specific baselines you would like us to compare against.
>
> W3: We agree that more complex simulators are used in practice, but we prioritised clear presentation by focusing on experiments that are easy to understand, familiar to the social simulation community, & complex enough to demonstrate our method's viability. As in our response to Q1: applying our method to more complex models is complicated by a lack of standard benchmarks in the social simulation community.
>
> L1: We agree that selecting the appropriate granularity via $\tau$ is hard, but in many cases domain experts know what emergent properties interest them. For instance, epidemiologists/labour economists/central bankers often care about the number infected individuals/unemployment numbers/inflation rates over time. In other words, domain expertise can be leveraged to select a suitable $\tau$. We do not know of existing work that automatically selects the right granularity for abstraction; the closest we know of applies rate distortion theory to learn low dimensional representations of MAB and RL problems [R1.10-11] but do not have a causal flavour & focus on single downstream tasks.
>
> **Refs**
>
> In Official Comment. (Too few chars sorry!)

---

> ### Author Response · Authors · 2024-08-06
> **References in Rebuttal**
>
> **Refs**
>
> [R1.1] Geffner et al. "Deep end-to-end causal inference" NeurIPS Workshop on Causal Machine Learning for Real-World Impact (2022)
>
> [R1.2] Melistas et al. "Benchmarking counterfactual image generation" arXiv (2022)
>
> [R1.3] Mooij et al. "Distinguishing cause from effect using observational data: Methods and benchmarks" JMLR (2016)
>
> [R1.4] White "Maximum likelihood estimation of misspecified models" Econometrica (1982)
>
> [R1.5] Aglietti et al. "Causal Bayesian optimization"  AISTATS (2020)
>
> [R1.6] Lee and Bareinboim "Structural causal bandits: where to intervene?" NeurIPS (2018)
>
> [R1.7] Lattimore et al. "Causal bandits: Learning good interventions via causal inference" NeurIPS (2016)
>
> [R1.8] Bilodeau et al. "Adaptively exploiting d-separators with causal bandits" NeurIPS (2022)
>
> [R1.9] Angione et al. "Using machine learning as a surrogate model for agent-based simulations" Plos one (2022)
>
> [R1.10] Arumugam and Van Roy "Deciding what to learn: A rate-distortion approach" ICLR (2021)
>
> [R1.11] Arumugam and Van Roy "Deciding what to model: Value-equivalent sampling for reinforcement learning" NeurIPS (2022)

---

> > ### Author Response · Authors · 2024-08-14
> >
> > Thank you once again for your feedback – we hope we have been able to address your questions in our responses. If you believe our improvements warrant an increase in your score, we would be very grateful to see this reflected in your updated score ahead of the deadline in ~25 mins.

---

### Author Rebuttal · Authors · 2024-08-06

We thank all reviewers for their feedback. We have responded to each of your points individually & are happy to expand on anything in the discussion period. We address some common points here.

**Additional experiment**

A common recommendation was to test our method in another experiment. We have therefore designed and run a further experiment, and provide details & results here. We'll use the extra space & appendix to describe & present the results of this extra case study, & hope this addresses `fR4E`, `fUGP`, & `3hmr`'s requests to include a further example from a different policy setting.

In this extra case study, we consider a different policy scenario: reintroducing a species into an ecology, & simulating the ensuing population dynamics. Specifically, we slightly adapted a model from [G1]: we model an environment initially consisting of `grass`, `sheep`, & `wolves`, in which `grass` grows & is eaten by `sheep`, `sheep` eat `grass` & reproduce & get eaten by `wolves`, and `wolves` eat `sheep` & reproduce. The intervention we consider entails reintroducing a third animal species -- `bears`, which eat both `sheep` and `wolves`, & also reproduce -- whose population is originally zero but is made non-zero at some intervention time $t$. (We imagine that $t$ is the variable the policymaker wants to optimise here.) This additional simulator is suitable for similar reasons to the SIRS ABM: predator-prey population dynamics models as in [G1] are easy to understand & are familiar to the social simulation community, yet are complex enough to demonstrate our method's viability.

We simulate the interactions between these 4 species in a spatial model, in which members of each animal species move around the grid and interact with the other species. We are then interested in understanding how the reintroduction of the `bears` affects the overall population dynamics (i.e., the counts of each animal in each species, along with the quantity of `grass` (i.e. the natural resource sustaining life) over time). As in the epidemic case study, we consider the problem of learning interventionally consistent surrogates for this complex spatiotemporal simulator, & once again examine three possible approaches for constructing surrogate families:

- a family of deterministic mechanistic models based on a discrete-time Lotka-Volterra model of population dynamics [see, e.g., G2], where (analogously to the `LODE` surrogate family discussed in the epidemic case study) the underlying deterministic dynamics of the population dynamics model index a probability distribution at each time step (in this case, a Binomial distribution for each of the 4 species);
- an `LRNN` family, exactly mirroring the `LRNN` family considered in the epidemic case study presented already;
- and a third family considers a hybrid approach, where (as in the `LODERNN` family considered in the epidemic case study) we pass a recurrent network over the underlying Lotka-Volterra-type population dynamics model first before taking the output of the recurrent network to index the Binomial distributions for each of the four species.

A table for the results of this additional case study is shown in the pdf document accompanying the Global Rebuttal, where we see that the results are qualitatively very similar to the epidemic case study already presented: we see that training surrogates using our framework yields significant improvements in the surrogates' interventional consistency over observationally trained baselines, and that interventionally trained surrogates only see a minor decrease in performance on observational data compared to the drop in performance the observational surrogates see on interventional data.

**Discussion of results & building benchmarks**

Some reviewers (namely `fR4E`, `fUGP` & `3hmr`) also suggested we expand our discussion of the results and of what benchmarks should be established for causal surrogate modelling. To this end, we will specifically discuss one further approach to benchmarking (beyond the metrics appearing in Table 1) based on the performance of causal surrogates on downstream decision-making tasks. In the context of the epidemic case study presented, we will discuss and use as a benchmark metric the question of how well the different surrogates preserve the ordering of interventions (lockdown vs. no lockdown) in terms of the degree to which they reduce the number of infections over time. The SIRS ABM predicts that any lockdown is better than no lockdown at all, and we would like for our surrogates to preserve this property (i.e. that no lockdown in the surrogate is also worse than introducing any lockdown).

We have checked for this property in both the interventional & observational surrogates, and see that the latter often do not predict that no lockdown is the worst option in this respect, and in some cases mistakenly predict that no lockdown is the _best_ option (e.g., the observational `LRNN` predicts that no lockdown was the best option in 1 of 5 training repeats, and was not the worst option in all 5 of 5 training repeats). In contrast, none of the interventional surrogates predict that no lockdown is the best option, and only the interventional `LODE` model predicts that no lockdown is not the worst option (in only 2 out of 5 training repeats).

We will use some of the extra space & appendix to expand on the discussion of the results in this way, and use this as an example of benchmarks that can be established & used in the future literature on interventionally consistent surrogate modelling.

**Refs**

[G1] Wilensky and Reisman "Thinking like a wolf, a sheep, or a firefly: Learning biology through constructing and testing computational theories—an embodied modeling approach." Cognition and instruction 24.2 (2006): 171-209.

[G2] Sabo "Stochasticity, predator–prey dynamics, and trigger harvest of nonnative predators." Ecology 86.9 (2005): 2329-2343.

---

> ### Author Response · Authors · 2024-08-13
>
> Thank you all for your feedback during the review and discussion period. If there are any further points that you would like to discuss before the discussion period closes, please do let us know and we will be happy to respond.
>
> We are particularly cognizant of the extreme discrepancy between the positive scores and feedback we have received from all reviewers (7 from `fR4E`, 6s from `fUGP` and `3hmr`, positive comments from all reviewers including `x5Gt`) and the single outlying "Strong Reject" score from Reviewer `x5Gt`, whose concern is the relation of our work to a paper appearing at UAI '24 approx. 4 weeks ago (herein [R3.1]).
>
> We have already used the rebuttal and discussion to highlight the many substantive differences between our work and [R3.1]'s. We have shown that [R3.1] differs substantially in focus from our work, and is entirely unsuitable for the settings we consider, given that it scales extremely poorly to the large-scale complex simulators of interest to us. We have also carefully consulted [R3.1]'s paper and used [R3.1]'s accompanying code to run computational experiments that verify these claims. (Please see the discussion below for details.)
>
> While a "Strong Reject" is, therefore, inappropriately low and incompatible with Reviewer `x5Gt`'s review and the ensuing discussion*, we would in particular like to invite any further comments from the reviewers on any concerns they may have in relation to [R3.1], in order that we may address them. And if you have no further concerns about our work, please do champion our paper!
>
> Thank you once again for your helpful feedback.
>
> *A "Strong Reject" is reserved for papers "with major technical flaws, and/or poor evaluation, limited impact, poor reproducibility and mostly unaddressed ethical considerations". In contrast, all reviews give "Soundness: 3: good" for our work, rather than claiming major technical flaws; "poor evaluation" is incompatible with the fact that we have provided two case studies, each studying 3 approaches to learning interventionally consistent surrogates, and conducted an experimental comparison against [R3.1]'s approach; "limited impact" is incompatible with the fact that **our work currently provides the only viable (practical, scalable, and interpretable) method to learn interventionally consistent surrogates for complex simulators** of the kind used to inform decision-making in complex systems; "poor reproducibility" is not raised by any reviewer; and no reviewers raise ethical concerns.

---

### Decision · Program_Chairs · 2024-09-25

**Decision:**

Accept (poster)

**Comment:**

This paper introduces a robust framework for learning surrogate models that maintain consistency under interventions, specifically designed for large-scale complex simulations. After a thorough review of the content, I recommend accepting the paper for the following reasons. The key strength of this work lies in its innovative approach to ensuring that surrogate models behave consistently under various interventions, a critical requirement for accurate and reliable simulation in policy-making and other high-stakes environments. The authors build upon established causal abstraction theories and provide strong theoretical foundations that justify their method. Their approach is particularly noteworthy for its ability to scale to large, complex systems, which are common in real-world applications but challenging for existing methods. The empirical validation, conducted on an epidemiological model, effectively demonstrates the method’s practical utility. The surrogate models trained with the proposed framework showed significant improvements in consistency under intervention, reducing the risk of erroneous policy decisions based on simulation results. The authors also address potential concerns about the method’s applicability to other domains by discussing extensions and providing a new case study in the rebuttal. Importantly, the criticism raised by Reviewer x5Gt regarding similarities with prior work is not a valid basis for rejection. The authors have convincingly demonstrated that their approach is substantially different from and more general than the work in question, particularly in its application to more complex, multi-variable systems and its scalability. This distinction highlights the novelty and impact of the paper, making it a significant contribution to the field. Given the strong theoretical contributions, empirical validation, and the clear distinction from contemporaneous work, this paper is a valuable addition to the literature and should be accepted.